# Resolving subcellular pH with a quantitative fluorescent lifetime biosensor

Joshua J. Rennick [1], Cameron J. Nowell [1], Colin W. Pouton [1] & Angus P. R. Johnston [1]✉

Changes in sub-cellular pH play a key role in metabolism, membrane transport, and triggering cargo release from therapeutic delivery systems. Most methods to measure pH rely on intensity changes of pH sensitive fluorophores, however, these measurements are hampered by high uncertainty in the inferred pH and the need for multiple fluorophores. To address this, here we combine pH dependant fluorescent lifetime imaging microscopy (pHLIM) with deep learning to accurately quantify sub-cellular pH in individual vesicles. We engineer the pH sensitive protein mApple to localise in the cytosol, endosomes, and lysosomes, and demonstrate that pHLIM can rapidly detect pH changes induced by drugs such as bafilomycin A1 and chloroquine. We also demonstrate that polyethylenimine (a common transfection reagent) does not exhibit a proton sponge effect and had no measurable impact on the pH of endocytic vesicles. pHLIM is a simple and quantitative method that will help to understand drug action and disease progression.

The regulation of pH and establishing pH gradients across cell membranes plays a crucial role in many cell functions, including membrane transport, energy production and degradation pathways[1]. Compartmentalisation of the eukaryotic cell into different organelles, each with specific chemical and pH environments, means the pH within a cell can range from <pH 5 in lysosomes[2] to ~pH 8 in the mitochondrial matrix[3]. Intracellular pH can be a marker of overall cell health, with apoptosis[4] and some disease states[5,6] displaying altered or dysregulated pH. Of particular interest is the pH gradient established in the endo/lysosomal pathway. The acidic environment within the lysosomes is necessary for the proper function of numerous enzymes essential for key physiological actions such as degradation of macromolecules and pathogens[7]. Genetic disorders that prevent lysosomal acidification can significantly affect homeostasis by causing a build-up of material destined for degradation[8]. This natural pH gradient is also exploited for drug delivery to enhance delivery of drug payloads into cells, either by triggering release of the drug from a carrier or disrupting the membranes in a pH-dependent manner to allow delivery into the cytosol[9–13]. Therefore, accurately measuring intracellular pH is vital to improve understanding of diseases while also helping in the development of potential treatments. Discerning changes in the pH of sub-cellular compartments in response to different stimuli, or due to dysregulation in the disease state, requires the use of quantitative tools to measure intracellular pH. However, current methods to measure intracellular pH have several limitations.

Numerous synthetic fluorophores have been engineered to change their fluorescence intensity in response to pH[14–16]. A substantial limitation of small molecule pH sensors is controlling where they localise inside the cell. Typically, these dyes are taken up into endosomal/lysosomal vesicles, but a number of these dyes are released from the endosomal compartment when they change their protonation state[17]. Therefore, it is challenging to ensure the pH that is measured comes from a specific organelle. To overcome this, genetically encodable pH-sensitive proteins can be used as an alternative. These protein-based pH sensors[18–22] can be fused to proteins that natively localise to specific organelles within the cell, ensuring the pH measurement comes from the desired location. They also have the benefit of exhibiting lower toxicity than their small molecule counterparts[17]. Most pH biosensors (protein or synthetic) measure a change in fluorescence intensity based on a change in the protonation state of the fluorophore. A limitation with all intensity-based pH sensors is decoupling the pH measurement from the concentration of the sensor. To distinguish between a high concentration of sensor with a low fluorescent signal from a low concentration of sensor with a high

[1]Monash Institute of Pharmaceutical Sciences, Monash University, Parkville, Victoria, Australia. ✉e-mail: angus.johnston@monash.edu

fluorescent signal, the pH dependant signal is typically referenced to a second pH-insensitive fluorophore. However, the necessity for the second fluorophore complicates the pH measurement through a combination of increased complexity of the sensor, FRET interactions between the fluorophores, and spectral overlap in the emission channels. Furthermore, an inherent limitation with intensity-based measurements is the drop in signal-to-noise ratio when the intensity of the pH responsive fluorophore decreases. The absolute error in each fluorescence measurement remains constant across the physiological pH range, but if the fluorescence intensity decreases at lower pH, the relative error increases substantially as the intensity approaches zero. This results in a high degree of uncertainty in the subsequent pH measurement. In addition to this, these sensors rely on interpolation from a sigmoidal curve, which has greater error in the exponential and asymptotic regions than linear regression. Both of these factors limit

the useful range of the sensors to a narrow pH band, which is typically smaller than relevant physiological pH values. Many of these challenges can be overcome by using fluorescent lifetime to infer pH[20,23,24], as it is independent of fluorophore concentration and can quantify pH without the need for a second reference fluorophore. However, pH measurements using fluorescence lifetime have previously lacked the spatial resolution necessary to ascertain the pH of individual subcellular compartments with high accuracy. To overcome the limitations with existing intracellular pH measurements, here we present a pH-dependent fluorescence lifetime imaging microscopy (pHLIM) approach to quantitatively determine intracellular pH (Fig. 1a, b). pHLIM uses the fluorescent protein mApple, which we have observed has a pH-dependent fluorescent lifetime that is linear across the physiologically relevant pH range. By expressing mApple as a fusion protein and developing an automated deep learning analysis tool, we can

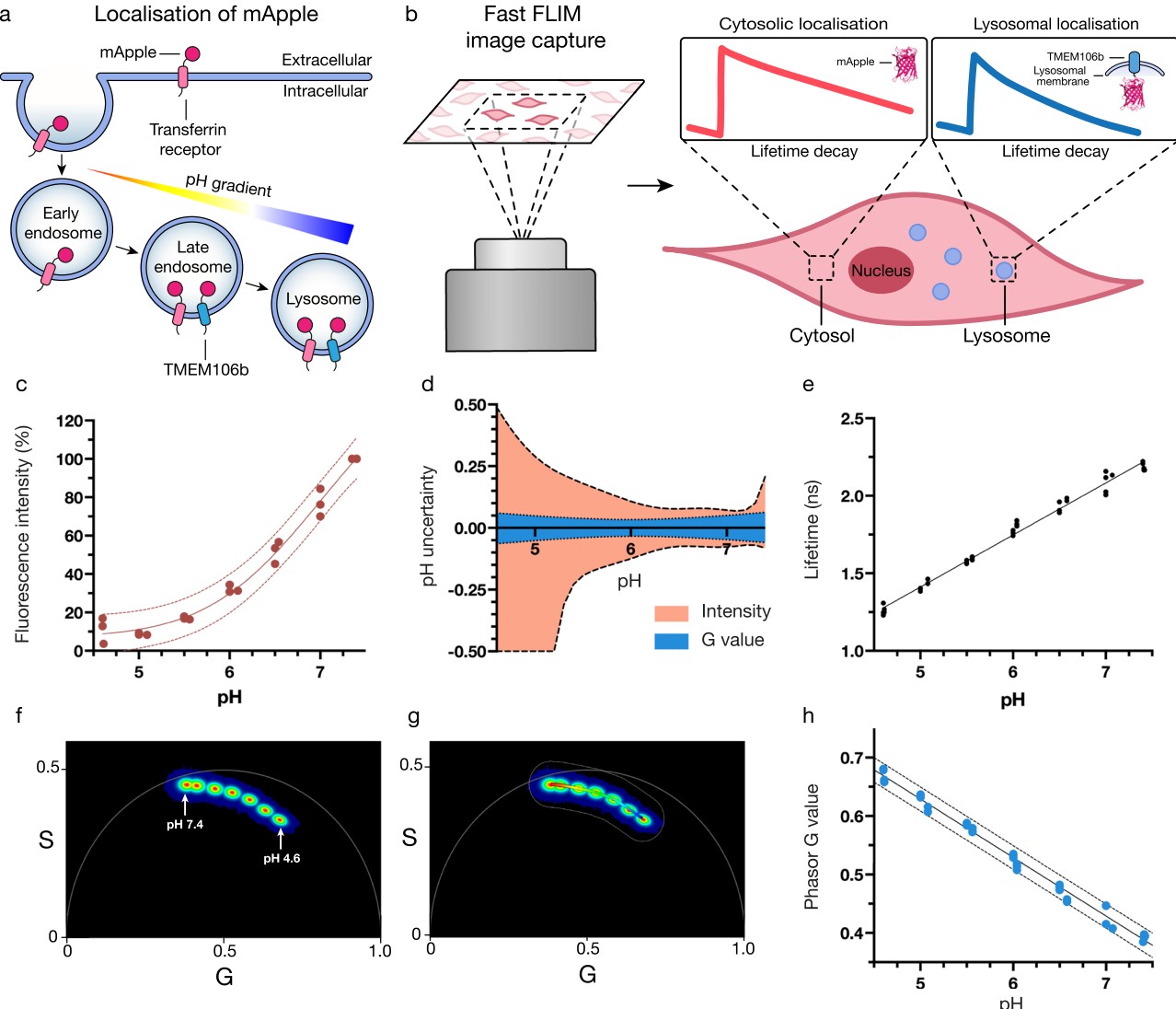

**Fig. 1 | mApple is a genetically encodable biosensor that can quantitatively determine subcellular pH using fast FLIM. a** Schematic of cellular membrane and endocytic vesicles with mApple fused to different transmembrane proteins to achieve targeted cellular localisation. pH gradient indicates increasing acidification as endosomes mature. Not to scale. **b** Overview of FLIM technique involving confocal microscopy of mApple expressing cells and subsequent analysis indicating the altered mApple fluorescence lifetime in different subcellular pH environments. **c** mApple fluorescence emission intensity over the calibration range with the least squares fit (solid line) and 95% prediction band (dotted lines). The points shown are the mean value from each of three independent experiments (*n* = 3). **d** Uncertainty

of interpolated pH as a function of actual pH for intensity or lifetime (G value) measurements, as determined by interpolation of (**c**), (**h**). **e** Calibration of recombinant mApple mean weighted fluorescent lifetime from pH 4.6–7.4 (*n* = 3). **f** pH dependence of recombinant mApple fluorescent lifetime visualised on a phasor plot, colour is indicative of the frequency of photons (red = high, blue = low), *n* = 3. **g** Equivalent phasor plot (**f**) with a 'phasor mask' applied which creates a pseudo colour scale that can be applied to fast FLIM confocal images (*n* = 3). **h** Extracted mean weighted G values from the phasor calibration, with a linear trendline (solid line) and 95% prediction band (dotted lines). The points shown are the mean value from each of three independent experiments (*n* = 3).

accurately determine the pH of different sub-cellular compartments in live cells in real time.

## Results

### Photophysical properties of mApple

mApple is an engineered protein that has been evolved from dsRed, a fluorescent protein originally isolated from a coral anemone (*Discosoma sp.*)[25,26]. It is part of a family of pH-sensitive fluorophores that includes pHuji[19], which was specifically evolved from mApple for use as an intensity-based pH biosensor. To measure the photophysical properties of mApple, we expressed and purified recombinant mApple from *E. coli* (Fig. S1). The pH dependence of mApple fluorescence was measured from pH 7.4–4.6 and we observed a 90% drop in the emission intensity across this range (Fig. 1c). The correlation between fluorescence intensity and pH showed a similar trend to other pH-sensitive proteins[18,19]. The principal limitation with using fluorescence intensity to determine pH (aside from the need for a reference fluorophore) is the increased uncertainty in the fluorescence measurement as the signal decreases. This is highlighted by plotting the uncertainty of the interpolated pH vs buffer pH (Fig. 1d), which shows uncertainty in the interpolated pH increased from ~0.2 at pH 7 to >1 at pH 4.6. This high degree of uncertainty in the intensity measurement limits the useful range of intensity-based biosensors.

To investigate the pH-responsive fluorescence lifetime of mApple, we used fast FLIM to calculate the fluorescent lifetime of mApple at each pH (Fig. 1e). We determined mApple lifetime to be 2.2 ns at pH 7.4, which is shorter than the 2.9 ns lifetime previously reported for mApple[27]. When the pH was decreased to 4.6 the lifetime decreased significantly to 1.3 ns. The change in lifetime followed a linear trend through the physiologically relevant pH range, with a lifetime change of ~0.34 ns per pH unit. This linear trend is in contrast to other lifetime sensitive fluorescent proteins (E2GFP[23] and ECFP[24]), which exhibit sigmoidal behaviour over the physiological pH range. Non-linear analysis complicates data fitting and lowers certainty in pH measurement at high and low pH. Although directly using the lifetime can be useful, detailed modelling of fluorescence lifetime can be complex as it requires fitting of several exponentials. To simplify our approach, we opted for the fit-free phasor analysis approach developed by Jameson, Gratton and Hall[28] which obviated the need for complex curve fitting. Using this method, lifetime data is displayed as a graphical representation on a phasor plot to indicate different species visually, by determining the sine (S) and cosine (G) Fourier transformations of normalised emission decays on a per pixel basis. Simplistically, each pixel in the image is correlated to a cartesian coordinate on the phasor plot, permitting the inference of pH without multi-exponential fitting. An overlay of phasor plots obtained for mApple indicated a narrow distribution at each of the tested pH values (Fig. 1f) and served as a calibration of the pH colour scale (Fig. 1g) used for image analysis later, herein denoted as a phasor mask. Analysis of the average G value within the pH range also displayed a linear trend (Fig. 1h), enabling simple and accurate determination of pH, with each pixel estimated to within 0.1 pH unit (Fig. 1d) using this G value calibration.

The linear dependence of mApple lifetime (G value) across the physiological pH range is in contrast to other commonly employed fluorescent proteins. muGFP[29] shows a similar drop in fluorescence intensity from pH 7.4 to 4.6 (~90%), however, the G value remains consistent across this pH range (Fig. S2a). When calibrating fluorescence intensity with pH, the signal-to-noise ratio decreases as the pH drops, leading to a significant increase in the uncertainty of the intensity measurement (Fig S3a). pHlourin[18], a fluorescent protein developed specifically as a pH sensor exhibits a drop in intensity from pH 7.6 to 6, but is not sensitive to pH below this range. Similar to muGFP, the uncertainty in the intensity measurement increases greatly as the pH decreases (Fig S3c). The G value of pHlourin shows some

dependence on pH between 7.6 to 6 (decreasing from G = 0.55 at pH 7.6 to G = 0.45 at pH 6), however, the magnitude of the lifetime change and the range over which the change occurs is less than observed for mApple (Fig. S2b). The fluorescence intensity of mCherry[30] is largely insensitive to pH (with a 20% drop in intensity below pH 5.5) and has no change in the G value across the physiological range (Fig. S2c). Each of these points highlight the advantages of using mApple as a pH biosensor.

This lifetime-based pH biosensor is a substantial improvement compared to intensity measurements of mApple, especially at pH below 6.5. Importantly, fluorescence lifetime is independent of protein concentration (Fig. S4a), which means fluorescence lifetime can be used as a single measurement without the need for a reference fluorophore to determine pH. Fluorescence lifetime is also independent of the ionic strength (Fig. S4b), which means variations of salt concentrations in different cellular organelles will not influence the pH measurement. Furthermore, the pH-induced lifetime change is reversible (Fig. S5), enabling dynamic changes in pH to be measured. Together, this demonstrates mApple is an excellent biosensor to detect changes in pH within a physiologically relevant pH range.

### mApple as an organelle-specific pH sensor

We next moved to express mApple in mammalian cells to allow pH measurements in different sub-cellular compartments (Fig. 1a, b). When mApple is expressed in NIH-3T3 cells without fusing it to another protein, it is distributed throughout the cytosol and nucleus (Fig. 2a, Figs. S6, S7). To localise mApple to specific cellular compartments, it was fused to proteins that natively traffic to the compartment of interest. In this study, we chose to examine two fusion proteins that traffic differently within the endo/lysosomal pathway, transferrin receptor 1 (TfR) and transmembrane protein 106b (TMEM106b). TfR is a rapidly internalised surface receptor important for iron uptake, which has a high abundance within the early endosomal trafficking pathway and is constantly recycled back to the cell surface[31]. Fusing mApple to the C-terminus of TfR enables the pH of the cell surface and endosomal pathway to be measured. In contrast to TfR, TMEM106b is a late-stage endo/lysosomal protein with unknown function. The C-terminus has been demonstrated to reside in the luminal domain of vesicle membranes and represents a method to assess the pH of late endosomes and lysosomes[32,33]. mApple fusions of both TfR (TfR-mApple) and TMEM106b (TMEM106b-mApple) were expressed in NIH-3T3 cells (Fig. 2b, c) and the localisation was confirmed by co-expression with mEmerald fused Rab5a (early endosome) or LAMP1 (lysosome) (Figs. S8, S9). Analysis of the confocal fluorescent images revealed the majority of TfR-mApple resides in vesicles inside the cell, whilst a proportion can be observed on the surface of the cell. Colocalisation of TfR-mApple was observed with Rab5a and LAMP1, indicating its presence throughout the endosomal trafficking pathway (Fig. S8a, b). The high prevalence of TfR-mApple in endosomal vesicles was expected due to the rapid turnover of TfR on the plasma membrane. TMEM106b-mApple was not observed on the plasma membrane and there was minimal colocalisation with Rab5a (Fig. S8c). However, substantial colocalisation was observed with LAMP1 (Fig. S8d), confirming its presence in lysosomes and indicating that TMEM106b-mApple is a good marker for the later stages of the endo/lysosomal pathway.

After confirming their localisation, we next analysed the fluorescent lifetime and phasor plot of each fusion protein to provide a broad pH overview. Applying a pH calibrated phasor mask to the confocal images (Fig. 2a–c) aids the visualisation of intracellular pH (Fig. 2d–f), while the phasor plot (Fig. 2g–i) indicates the distribution of measured pH values. The cytosolically expressed mApple formed a tight population on the phasor plot (Fig. 2g), and correspondingly showed a homogenous red colour throughout the cell in the phasor mask image (Fig. 2d). A mean G value of 0.39 was obtained from the

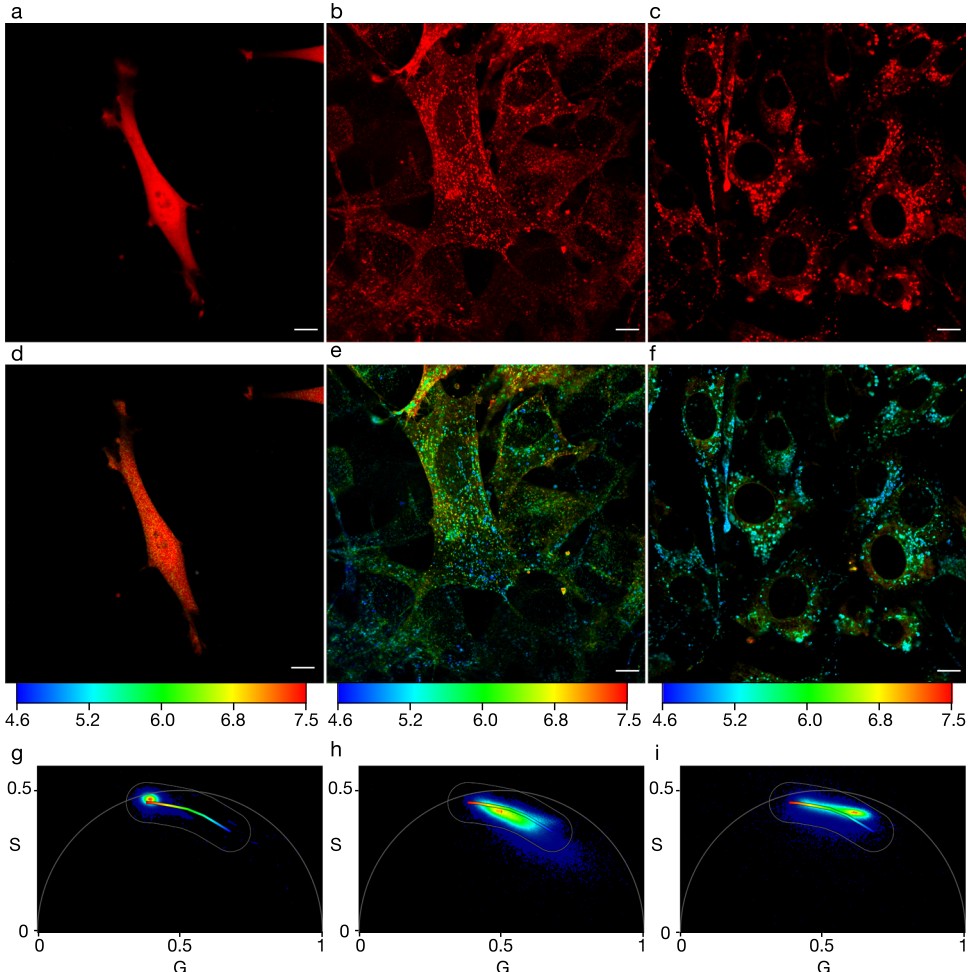

**Fig. 2 | Fusing mApple to localisation tags enables the pH of different sub-cellular compartments to be visualised. a–c** Confocal images of NIH-3T3 cells expressing cytosolic mApple (**a**), TfR-mApple (**b**), and TMEM106b-mApple (**c**), pseudo-coloured red. **d–f** Corresponding fast FLIM images of **a–c** pseudo-coloured according to their pH. pH colour scale indicated underneath images. **g**, **h**, **i** Corresponding phasor plots of (**d**), (**e**), (**f**), respectively, with an overlayed phasor mask. Phasor plot colour is indicative of the frequency of photons at that phasor position (red = high, blue = low). Scale bar = 10 μm.

phasor plot for this image, which corresponds to a cytosolic pH of 7.4, based on the G value calibration (Fig. 1g).

The phasor overlayed image and phasor plot of TfR-mApple (Fig. 2e, h) showed a markedly different pattern to cytosolic mApple. From the phasor plot, the mean G value of TfR-mApple was 0.52, which corresponds to a pH of 6.1, and the modal G value was 0.45 which corresponded to a pH of 6.8. Unlike cytosolic mApple where there was a tight distribution of pH, the TfR-mApple image exhibited a broad pH range, with the majority (majority is calculated as the 0.125 and 0.875 quantiles (middle 75%) of the weighted mean G value which is then converted to pH using the calibration from Fig. 1h.) of pH values falling between 5.2 and 7.0. The broad pH range was anticipated due to the presence of TfR-mApple on the cell surface and in endo/lysosomal vesicles. These different populations of TfR-mApple (high pH surface and early endosome, and low pH late endosome/lysosome) can be easily distinguished by the lifetime measurements (Fig. S10).

TMEM106b-mApple also displayed punctate fluorescence (Fig. 2c, f) consistent with endo/lysosomal localisation and accordingly the phasor plot was skewed to higher G values (lower pH– Fig. 2i) compared to both cytosolic and TfR localised mApple. TMEM106b-mApple also displayed low levels of cytosolic fluorescence, likely due to over-expression of the protein. This signal was substantially lower intensity than the signal from the endosomes/lysosomes and if a low intensity (15 photons/pixel) threshold is applied to the images, the cytosolic signal can be removed. The mean

G value for the TMEM106b-mApple image was 0.57, which corresponds to a pH of 5.6, while the modal G value was 0.62 corresponding to a pH of 5.1. Both mean and modal pH was lower than TfR-mApple, again indicating that TMEM106b-mApple predominantly resides later in the endo/lysosomal trafficking pathway. TMEM106b-mApple exhibited a similarly broad pH range, with the majority between the pH of 4.8 and 6.8.

### Quantification of sub-cellular pH

Although analysing the phasor plot of several whole cells can be useful to give a qualitative picture of pH diversity in the cell, the wide distribution of pH measurements observed for TMEM106b and TfR-mApple highlights the heterogeneous nature of pH environments within the cell. FLIM provides spatial resolution down to the limits of confocal microscopy (~250 nm), which enables analysis of individual endosomes. We have demonstrated the high spatial resolution of the data with the pH of individual endo/lysosomes clearly visible (Fig. 2e, f). These images also show that the wide pH distribution observed stems from individual endosomes with different pH, rather than uncertainty in the pH measurement (Fig. S11).

Although it is possible to manually mask and measure the pH of individual endo/lysosomes, this type of image analysis by its nature is subjective and is time consuming due to the large number of vesicles detected per cell (>50). To address this, we have trained an established convolutional neural network (StarDist)[34] to identify vesicles and

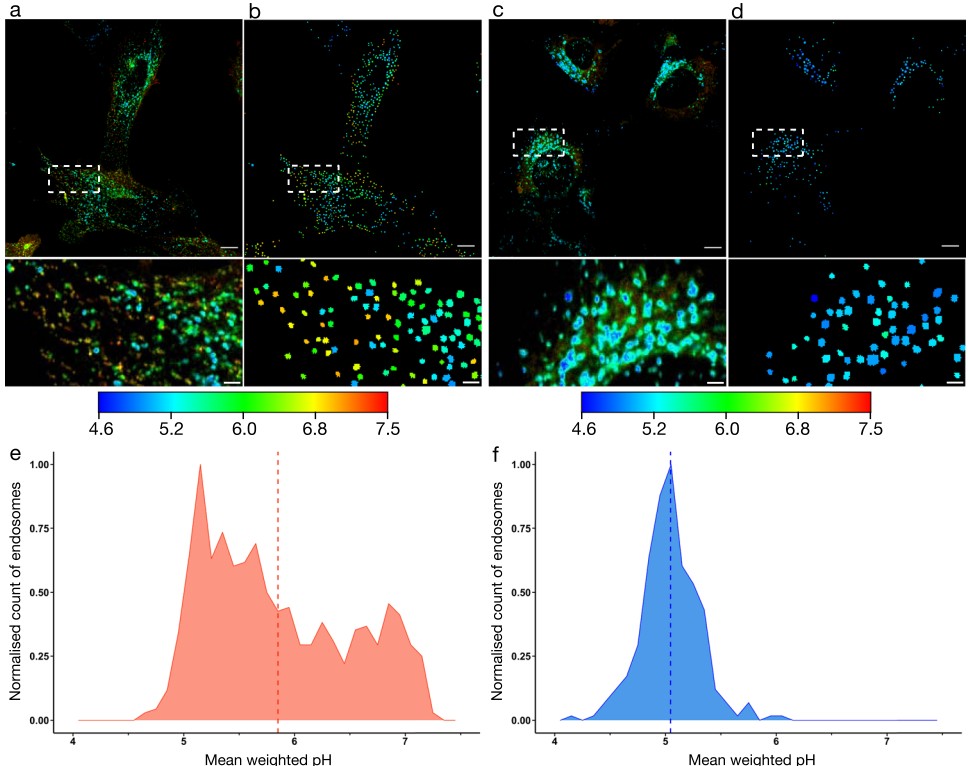

**Fig. 3 | Employing an automated deep learning model to detect endo/lysosomal compartments enables quantification of the intracellular pH distribution.** **a**, **c** Fluorescence microscopy images of NIH-3T3 cells expressing TfR-mApple (**a**) or TMEM106b-mApple (**c**). Images pseudo coloured by a phasor mask (Fig. 1g) according to the colour scale shown. **b**, **d** Automated detection of endo/lysosomes in (**a**), (**c**), respectively, pseudo coloured according to the indicated colour scale. Colour shown is indicative of the pH of the detected endosome. Zoomed insets are shown below each image. Scale bar = 10 μm, inset scale bar = 2 μm. **e**, **f** Histograms of TfR-mApple (**e**) and TMEM106b-mApple (**f**) showing the distribution of endo/lysosomal pH from (**b**), (**d**), respectively. The pH of each individual endosome is plotted in the histogram with the mean pH of the population shown as a dotted line.

developed an algorithm to determine the pH of each detected endosome. This algorithm calculates the endosomal pH by interrogating each pixel in the detected endosome and calculating the intensity weighted mean G value (Eq. (1)).

$$\text{Mean weighted G value (vesicle)} = \frac{\sum(\text{Pixel photon count} \times \text{Pixel G value})}{\sum \text{Photon counts in vesicle}}$$
(1)

The G value is converted to pH using the calibration curve in Fig. 1h. Using this workflow, we analysed the pH of >3700 individual vesicles from >30 TfR-mApple cells and >6000 individual vesicles from >150 TMEM-mApple cells (Fig. 3, Fig. S12). The histograms show the pH of individual vesicles and highlight the variation in pH between individual vesicles in the cell. The ability to observe the distribution of pH within the cell is a key advantage of the pHLIM technique. These results show a similar trend to that observed in the bulk analysis above, but enable pH quantitation of individual vesicles within the cell. In NIH-3T3 cells, TfR-mApple endosomes exhibited a higher mean pH (5.9) than the TMEM106b-mApple endosomes (5.0) (Fig. 3e, f). Similar results were observed in HEK293 cells (Figs. S13, S14). The TfR-mApple mean pH was significantly reduced in this analysis compared to manual analysis of the whole image, as the algorithm limits the contribution of the surface signal, which otherwise skews the mean TfR-mApple pH higher. The analysis of individual endosomes shows a narrower distribution of pH for TMEM106b-mApple vesicles (majority between 4.8 and 5.3) compared to the TfR-mApple endosomes (majority between 5.1 and 6.8). Similar results were observed when the analysis was expanded to further images (Fig. S12). We also investigated the intra-vesicular pH variability of the detected compartments (Fig. 2b, c),

which indicated that the average standard deviation in pH within each vesicle for TfR-mApple was 0.22, compared to 0.19 for TMEM-mApple (Fig. S11).

### Temporal resolution of pH
The dynamic nature of endosomal trafficking means that pH can change rapidly within the cell. To demonstrate the temporal resolution of this method, we acquired images at a rate of 0.7 frame/s to track dynamic changes in endosomal pH (Supplementary movie 1).

To demonstrate the mApple pHLIM sensor can be used to probe changes in pH induced by drugs, we next moved to assess the effect of adding the V-ATPase H⁺ pump inhibitor bafilomycin A1[35] (BafA1). We first confirmed that the presence of BafA1 did not influence the lifetime of mApple (Fig. S15). Incubating TMEM106b-mApple expressing cells with 100 nM BafA1 for 1 h resulted in a clear increase in pH, both visually by colour differences (Fig. 4a, b) and when the pH distribution of vesicles was plotted (Fig. 4c). BafA1 caused a significant ($p < 0.05$) increase in the total mean vesicle pH (Fig. 4d, Supplementary movie 2). The average pH of TMEM106b-mApple endosomes increased from pH 5.3 (G = 0.60) to pH 5.7 (G = 0.55) after 15 min, then to pH 6.1 (G = 0.52) after 30 min and pH 6.5 (G = 0.48) after 60 min. The addition of chloroquine also resulted in a similar increase in endosomal pH (Figs. S16, S17). This demonstrates the utility of the sensor to measure dynamic changes in pH over time within individual vesicles.

### Applying pHLIM to probe the buffering capacity of PEI
Finally, we used the mApple pHLIM sensor to probe the proposed proton sponge hypothesis, which has been suggested to help rupture endo/lysosomes and deliver therapeutic cargo to the cytosol[36].

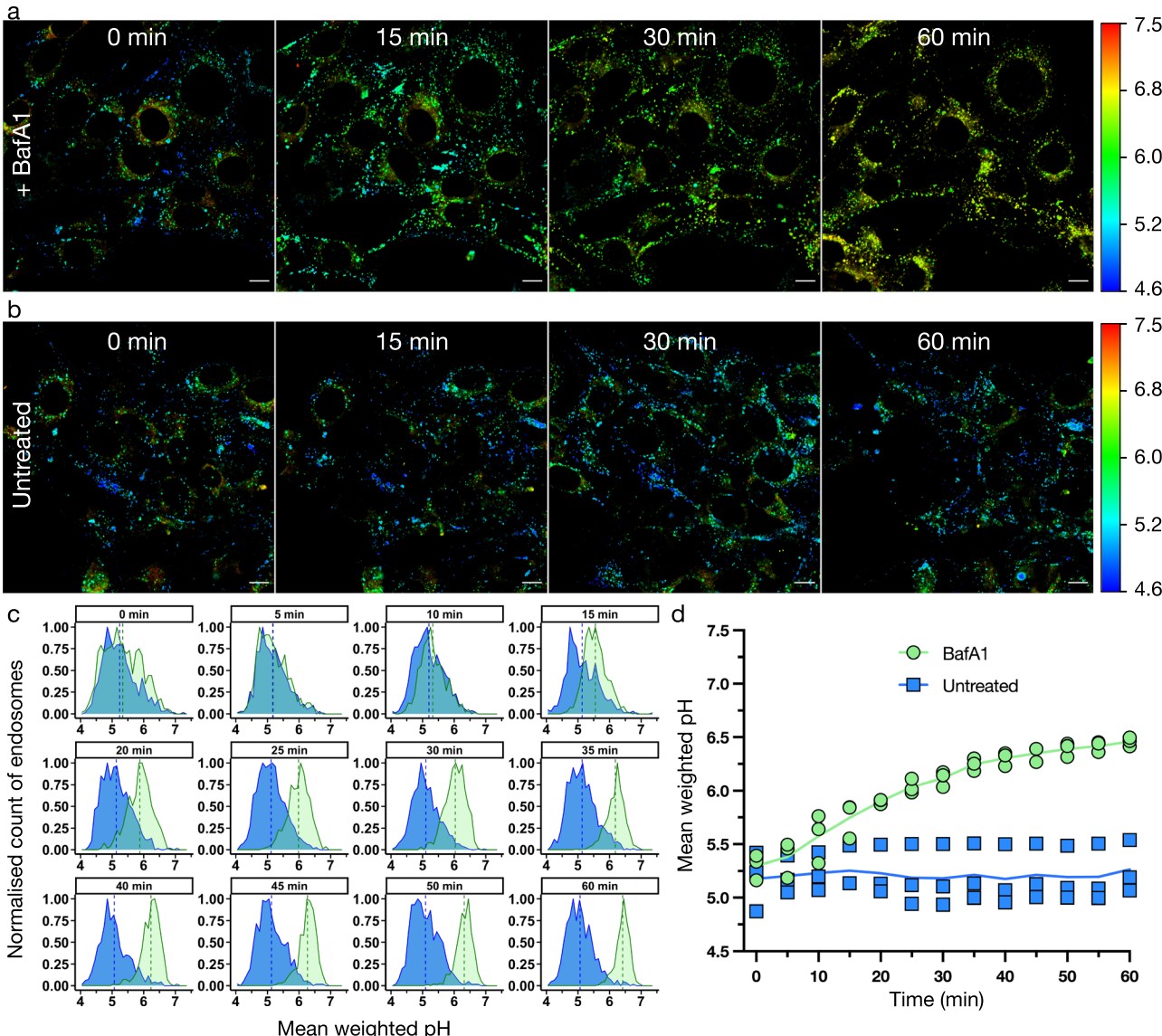

**Fig. 4 | mApple pHLIM sensor enables dynamic tracking of intracellular pH in response to treatment with bafilomycin A1. a, b** Time course fluorescence microscopy images of NIH-3T3 cells expressing TMEM106b-mApple treated with **a** 100 nM BafA1 or **b** untreated. Images pseudo coloured by a phasor mask (Fig. 1g), pH scale shown on right. **c** Histograms at the indicated timepoints showing the distribution of pH in endo/lysosomal compartments in a single image. Untreated (blue), BafA1 treated (green). The pH of each individual endosome is plotted in the histogram with the mean pH of the population shown as a dotted line. **d** Summary plot showing the mean pH of three replicate data sets, untreated (blue squares), BafA1 (green circles), solid line indicates the mean of three independent replicates. Scale bar = 10 μm, two-tailed unpaired student's *t*-test was used to analyse each time-point, * denotes *p* value <0.05 which applies to all timepoints from 15 min onwards. Time = 15 min, *p* = 0.033. Time = 60 min, *p* = 0.012 (*n* = 3).

We followed the uptake of Cy5 labelled polyethylenimine (PEI) over 6 h into TMEM106b-mApple transduced NIH-3T3 cells. Colocalisation of PEI with mApple was observed after 30 min (Figs. S18, S19). Over the same period of time, we investigated the effects of unlabelled PEI upon lysosomal pH. Phasor overlayed images did not reveal a population of intracellular vesicles with elevated pH after treatment with PEI (Fig. 5a, b) which was also verified when our automated algorithm was used to analyse vesicle pH in comparison to untreated samples (Fig. 5c). Over the 6-h time course, the mean pH of detected vesicles ranged from pH 5.0–5.3 for both the untreated and PEI treated samples (Fig. 5d). These results suggest that there was not an observable increase in lysosomal pH after treatment with PEI over 6 h.

To further investigate the potential buffering effect of PEI, we probed the pH of endosomal compartments containing PEI/DNA complexes. pDNA that encodes for EGFP was complexed with Cy5

labelled PEI. The polyplexes were incubated with TMEM106b-mApple transduced NIH-3T3 cells (2 μg/mL DNA concentration) for 4 or 6 h, and both incubation times resulted in strong GFP expression in ~50% of cells after 24 h (Fig. S20). Using the StarDist algorithm, we identified all the mApple positive vesicles as well as Cy5 positive vesicles that contain the PEI/DNA polyplexes (Fig. S21). We then measured the pH of the double Cy5/mApple positive endosomes and compared them to the pH of the Cy5 negative, mApple positive endosomes (Fig. 4e). Confirming the result observed for PEI by itself, the pH of endosomes containing PEI/DNA polyplexes was not significantly different to the pH of PEI/DNA negative endosomes (both ~pH 5.5– Fig. S22). We further investigated to see if there was a correlation between the amount of PEI in each endosome (measured from the Cy5 intensity) and the pH of the endosome. Increased sequestration of PEI in endosomes did not correlate with increased endosomal pH (Fig. 5f and Fig. S23).

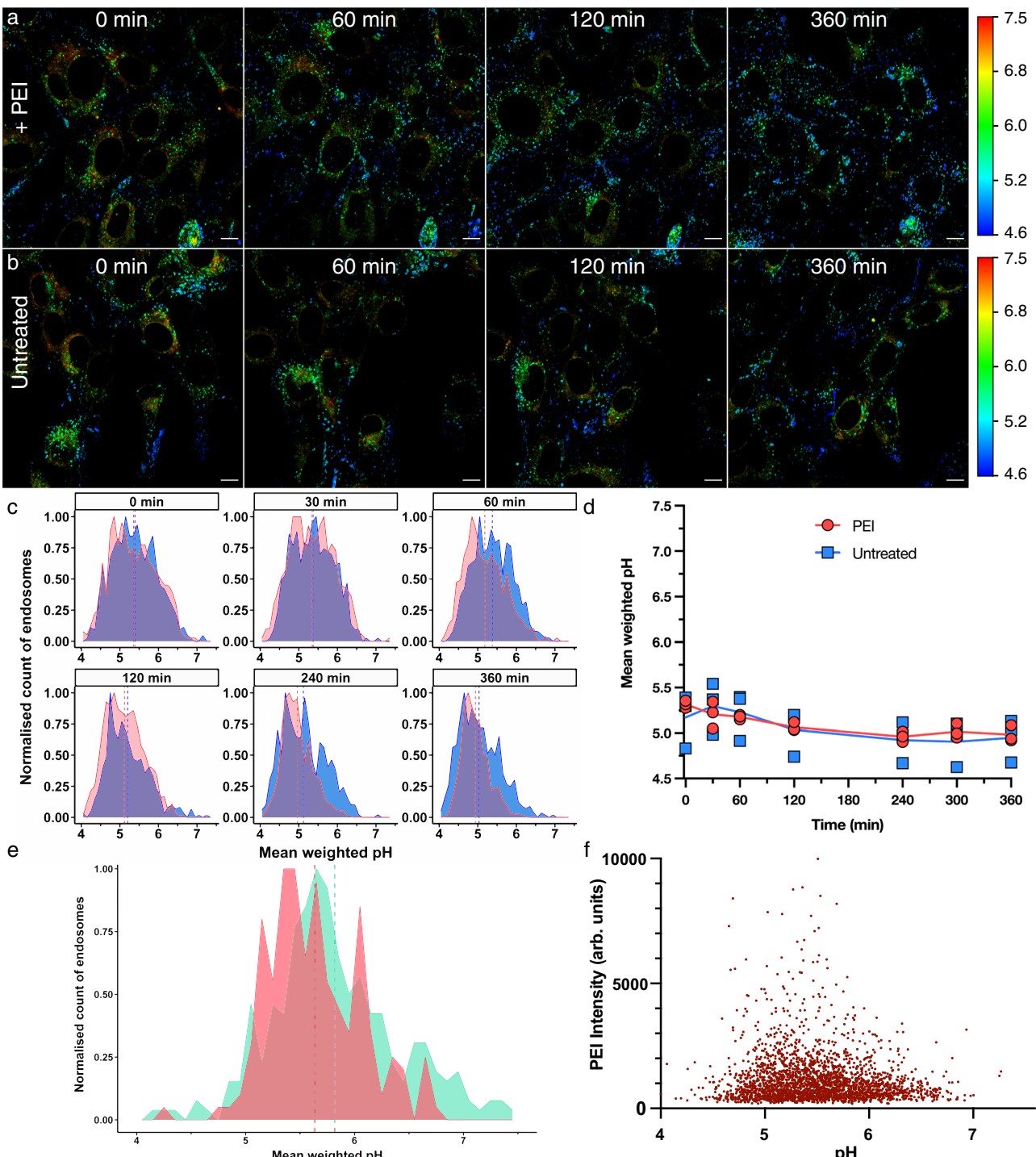

**Fig. 5 | Employing the mApple pHLIM sensor to probe for the purported proton sponge effect of PEI. a**, **b** Time course FLIM images of NIH-3T3 cells expressing TMEM106b-mApple treated with **a** 80 μg mL⁻¹ PEI or **b** untreated. Images pseudo coloured by a phasor mask (Fig. 1g), scale bar = 10 μm. **c** Histograms of untreated (blue), PEI treated (red) at the indicated timepoints showing the distribution of pH in endo/lysosomal compartments in a single image, with the mean pH shown as a dotted line. **d** Summary plot of the vesicle pH, untreated (blue squares), PEI (red circles), solid line indicates the mean of three independent replicates (*n* = 3).

**e** Histograms showing the pH distribution of vesicles in cells treated with pDNA/PEI complexes (2 μg/mL pDNA, 80 μg mL⁻¹ Cy5 labelled PEI) for 6 h. Cy5 signal was used to identify vesicles that contain PEI (red) and vesicles that do not contain PEI (teal). The pH of each individual endosome is plotted in the histogram with the mean pH of the population shown as a dotted line. **f** Correlation between PEI concentration (Cy5 signal) and vesicle pH for >2500 individual endosomes (from *n* = 3 experiments) identified to contain PEI (cells treated with pDNA/PEI complexes (2 μg/mL pDNA, 80 μg mL⁻¹ Cy5 labelled PEI) for 6 h).

## Discussion

Intensity-based pH measurements are hampered by the inherent drop in signal when the sensors are in their low-intensity state. Measuring the intensity of sensors when the signal has dropped to

<10% of the original signal (and sometimes <1%) results in elevated levels of uncertainty in the pH measurement. It is important to note that the uncertainty in the inferred pH comes from both the intensity/lifetime measurement, and the pH calibration. This latter source

of uncertainty is often ignored, but can significantly affect the accuracy of the measurement. When modelling the sigmoidal response of intensity to pH, the exponential and asymptotic regions of the curve have substantially higher uncertainty than the linear region. This is exemplified by the interpolation of pH from the intensity of muGFP, pHluorin and mApple (Fig S3). For each of these fluorophores, the uncertainty in pH is substantially higher than if the lifetime of mApple is used for pH calibration. This uncertainly can be >1 pH unit, which greatly limits the application of intensity measurements to determine physiologically relevant pH changes. Furthermore, the need for a reference fluorophore that is truly pH insensitive along with potential complications with FRET makes it difficult to ensure intensity-based measurements accurately reflect the true pH. mApple exhibits a large (1 ns), linear shift in its fluorescence lifetime in response to changes in environmental pH. The linear response of mApple fluorescent lifetime within the physiological pH range makes pH interpretation simpler and less prone to high levels of uncertainty compared to previously reported lifetime sensitive fluorophores[20,24]. Using the fluorescent lifetime decouples the pH measurement from the concentration of the sensor, eliminating the need for a reference fluorophore and associated ratiometric analysis. Despite mApple exhibiting reduced intensity at lower pH, the lifetime measurements have the same level of certainty across the physiological pH range, resulting in <0.1 pH uncertainty in the measurements on a per-pixel basis. The increased accuracy of the pH measurements using pHLIM means we can identify pH changes with a high degree of certainty as soon as subtle changes occur (as per treatment with BafA1–Fig. 4) and definitively demonstrate when no changes occur (as per treatment with PEI–Fig. 5).

When expressed as a fusion protein, mApple was able to determine the pH of different subcellular compartments. Unmodified mApple distributed throughout the cytosol and as expected measured a pH of 7.4. Fusing mApple to TfR enabled visualisation of high pH (-pH 7.4) on the cell surface, as well as a diverse range of lower pH (5.2–7.0) inside the cell, consistent with localisation in the endosomal and lysosomal pathway. In comparison, TMEM106b-mApple showed no surface signal and localised in low pH vesicles (e.g. lysosomes). Colocalisation analysis with Rab5a and LAMP1 confirmed these observations, with TfR-mApple colocalising with both Rab5a and LAMP1, but TMEM106b-mApple only localising with LAMP1.

To enable quantitative analysis of sub-cellular pH, we implemented a StarDist[34] deep learning algorithm to automatically detect intracellular vesicles. StarDist was trained to identify endosomal compartments using 8 images containing a total of 1086 individually segmented endosomes. This enabled the detection of the majority of intracellular vesicles, whilst minimising the occurrence of false positives (Fig. S24). Implementation of this algorithm permitted the pH analysis of several thousand individual endo/lysosomes across multiple images, which would otherwise be time consuming with manual analysis. This is a key advance over previous pH studies which lack the specific compartment labelling and spatial resolution to identify individual sub-cellular compartments. The automated analysis of each vesicle enables both quantitative determination of the pH, as well as determining the range and diversity of pH within the cell. In addition, it permits the dynamic measurement of labelled vesicles over several hours.

The ability to quantify sub-cellular pH is important for a range of applications. For example, the acidifying endo/lysosomal pathway poses as a significant opportunity to enhance the specificity of drug delivery[37]. Through employing pH-responsive systems such as nanomaterials[38] and linkers (e.g. acetal[11,13]), the acidifying environment can facilitate cargo delivery to desired pH compartments in the trafficking pathway. Therefore, visualising the pH of subcellular locations where these materials natively traffic is of keen interest to optimise the design of these systems.

pHLIM enabled us to make dynamic quantitative pH measurements and assess the effect of two different reagents (bafilomycin A1 and PEI) on endo/lysosomal pH. Bafilomycin A1 is a V-ATPase inhibitor that prevents endo/lysosomes from maintaining low pH and is widely used to study acidification of these vesicles[20,35]. We observed a rapid onset of BafA1's neutralising effects with the endo/lysosomes of treated cells becoming significantly ($p < 0.05$) higher pH than untreated cells after 15 min. The neutralising effects continued over the 1-h incubation, however, began to plateau after 50 minutes at a mean pH of 6.5 compared to untreated control at pH 5.3.

BafA1 increases endosomal pH by inhibiting the V-ATPase H$^+$ pump[35], which prevents acidification of vesicles. It has also been shown that BafA1 inhibits the SERCA Ca$^{2+}$ pump, which disrupts autophagosome/lysosomal fusion independently of its effect on lysosomal pH[39,40]. However, it is possible the disruption of lysosomal fusion could play a role in increasing the pH of endosomes. To investigate this further we analysed the distribution of vesicle pH and the number of vesicles detected throughout the BafA1 treatment. If V-ATPase H$^+$ pump inhibition is the primary mechanism for increasing the endosomal pH, we would expect to see a steady increase in the pH of all the endocytic vesicles. However, if disruption of autophagosome/lysosomal fusion prevents acidification of the vesicles, we would expect to observe two vesicle populations; the initial population of vesicles with lower pH; and a new population of vesicles with a higher pH that are unable to fuse to the autophagosome/lysosomal compartments. Conventional analysis of the average pH inside the cell would not be able to distinguish between these two mechanisms, as in both cases the overall pH of the cell would increase. However, by analysing the individual vesicles we observed a single pH distribution with increasing (and narrower) pH (Fig. 4c), with a similar number of vesicles present throughout the experiment (Fig. S25). This shows that the primary mechanism of BafA1-induced lysosomal neutralisation is inhibition of V-ATPase H$^+$ pumps. It should be noted that this result does not contradict the findings that BafA1 can also inhibit autophagosome/lysosomal fusion. However, the lack of a second population of vesicles with a higher pH and the consistent number of vesicles suggests that over the 60-min time course, inhibition of autophagosomal/lysosomal fusion is not the driving force behind neutralisation of TMEM106b + vesicles. This analysis highlights the usefulness of pHLIM in making dynamic intracellular measurements of all detected vesicles within the field of view.

We next investigated the purported buffering effects of PEI upon vesicular pH. The delivery of biological therapeutics to their site of action in the cytosol is a significant challenge, as most biologics which are endocytosed into these endo/lysosomal compartments are degraded[41]. Delivery to the cytosol (also referred to as endosomal escape[42]) is very inefficient, with <2% of internalised material being trafficked to the cytosol[43,44]. To overcome this, some pH-responsive materials can be engineered to induce endosomal escape in the endo/lysosomal pathway[45]. However, the mechanisms by which these materials escape the endosome is not clear and is hotly contested. One proposed mechanism is the proton sponge effect[36], where polymers can buffer the acidification of endosomal compartments, which in turn leads to an increase in osmotic pressure as counter ions are pumped into the endosomes to balance the overall charge. It is proposed the osmotic pressure reaches a point which ruptures the endosomal compartment, delivering the contents of the endosome to the cytosol. There is mounting evidence to suggest there are significant limitations with this hypothesis, including demonstrating that polymers with high buffering capacity do not have increased endosomal escape[46], and ratiometric pH studies have failed to observe buffering of the pH[47]. However, these ratiometric methods have been hampered by high levels of uncertainty in the inferred pH (as demonstrated in Fig. S3) and have typically relied on treating the cells with synthetic pH sensors that do not necessarily localise to the same cellular compartments as the

polymers. A number of reports also lack evidence to show colocalisation of PEI with the specific subcellular compartments that are being measured. Here, we have demonstrated that Cy5 labelled PEI strongly colocalises with TMEM106b-mApple within 60 min. Despite this strong colocalisation, we did not observe any buffering from the PEI over 6 h. This was in stark comparison to BafA1 where elevated pH effects were observed as early as 15 min after treatment. By using Cy5 labelled PEI/DNA complexes, we were also able to measure the pH of individual PEI/DNA positive endosomes and compare the pH to endosomes in the same cell without PEI/DNA (Fig. 5e). There was no difference in the mean pH or the pH distribution in TMEM106b-mApple vesicles with or without PEI. Furthermore, in addition to measuring the pH of each individual endosome, we measured the fluorescence intensity of Cy5 in each endosome to determine the relative amount of PEI. We would anticipate that if PEI exerts a buffering effect in endocytic vesicles, vesicles with a greater amount of PEI would have a higher pH. By plotting the amount of PEI vs pH for >2500 individual endosomes from 3 independent replicates (Fig. 5f) we have shown that increased sequestration of PEI in endosomes does not correspond to a higher endosomal pH. Our results here show that (a) the average pH of vesicles does not change with PEI treatment, (b) there is no population of vesicles with higher pH and the distribution of pH is similar regardless of if the vesicle contains PEI or not, (c) there is no correlation between the amount of PEI in the vesicle and the pH. All combined, this strongly suggests that the proton sponge effect is not the predominant mechanism by which cytosolic delivery is induced by PEI.

We have demonstrated that FLIM measurements of mApple, combined with automated analysis of individual endosomes enables quantitative and accurate measurement of intracellular pH across the physiologically relevant pH range. This technique has a number of advantages over existing methods. (1) Simplicity: FLIM only requires a single measurement, rather than needing ratiometric measurements of two fluorophores. (2) Accuracy: our pHLIM measurements are accurate to <0.1 pH unit, compared to >0.5 for intensity-based measurements. (3) Responsive range: mApple exhibits a linear lifetime response across the tested physiological pH range. (4) Sub-cellular quantification: the application of StarDist enables the distribution of pH within the cell to be determined. (5) Endosome composition: we can identify which endosomes contain material (such as PEI) and correlate the pH to the amount of material in the endosome. Furthermore, because mApple is a genetically encodable sensor, we were able to express it in various intracellular locations such as the cytosol, cell surface, endosomes or lysosomes, which permitted local pH measurements at each of these locations. We were able to interrogate pH changes in response to treatment with bafilomycin A1 and PEI. Although substantial changes in lysosomal pH were observed with BafA1, changes in lysosomal pH were not observed over 6 h despite substantial colocalisation of PEI with these compartments. These results highlight the power of coupling a genetically encodable pH sensor with an automated detection and analysis workflow to make robust intracellular pH measurements. The simple and quantitative pHLIM technique outlined here has the potential to improve our understanding of drug action in addition to disease progression and will also be a valuable tool to help design the next generation of controlled drug release systems.

## Methods
### Buffers and materials
All chemicals and materials were purchased from Sigma Aldrich except where specified. Cell culture materials were purchased from ThermoFisher Scientific. DNA cloning reagents including restriction enzymes, DNA polymerases and NEBuilder HiFi DNA assembly master mix were obtained from New England Biolabs. Buffers for assessing fluorescence lifetime were composed of either 0.01 M PBS (pH 6.5–7.4) or a 0.01 M citrate buffer (pH 4.6–6.0).

### Cell culture
NIH-3T3 (ATCC: CRL-1658), HEK293 (ATCC: CRL-1573) and HEK293-FT (HEK, ThermoFisher Scientific R70007) were maintained in Dulbecco's modified Eagle medium (DMEM), high glucose (GlutaMAX) with phenol red and 20% (NIH-3T3) or 10% (HEK293, HEK293-FT) foetal bovine serum (FBS) and 1% penicillin/streptomycin at 37 °C with 5% $CO_2$. NIH-3T3 media was supplemented with 2 μg/mL puromycin to maintain positive integrants. No authentication was performed as the cells were obtained from a reliable source. All cell lines were tested monthly for mycoplasma contamination by PCR. All cell lines were negative for mycoplasma.

### Plasmid construction
All plasmids were constructed using NEBuilder HiFi DNA assembly master mix with PCR products, vector restriction digests or DNA oligonucleotides with compatible overhangs. All synthetic oligonucleotides were obtained from Integrated DNA Technologies (IDT). Cloning was performed in TOP10 chemically competent *Escherichia coli* (*E. coli*) (ThermoFisher Scientific). mApple and TfR DNA were obtained from mApple-Lysosomes-20 (RRID:Addgene_54921) and mCherry-TFR-20 (RRID:Addgene_55144), respectively, which were a gift from Michael Davidson. The sequence for transmembrane protein 106b (TMEM106b) was obtained from the Gene database of the National Center for Biotechnology Information[48] (Gene ID: 54664) and ordered as a plasmid from Twist Bioscience, inserted into pTwist Lenti SFFV Puro WPRE. TMEM106b and TfR DNA were amplified for insertion as an N-terminal fusion to mApple and subcloned into the third-generation lentiviral plasmid pCDH-EF1-IRES-Puro (System Biosciences) which was digested with EcoRI and NotI (TMEM106b) or NheI and NotI (TfR) restriction enzymes. mApple was also inserted into pCDH-EF1-IRES-Puro alone by amplifying the mApple DNA from mApple-Lysosomes-20. These plasmids are available from Addgene (RRID:Addgene 179383, 179384 and 179385). The plasmid encoding CMV-EGFP was generated by digesting sfGFP-TFR-20 (RRID: Addgene_56488, a gift from Michael Davidson) with NheI and AgeI restriction enzymes before blunting with T4 DNA Polymerase, and blunt end ligation with T4 DNA ligase. For expression in *E. coli*, mApple DNA was inserted into pET His6 TEV LIC cloning vector (1B), a gift from Scott Gradia (RRID:Addgene_29653). mEmerald-Rab5a and mEmerald-Lysosomes-20 were both gifts from Michael Davidson (RRID:Addgene_54243 and RRID:Addgene_54149, respectively).

### Protein expression and purification
pET-His6-mApple was recombinantly expressed and purified using a previously reported method[44] by transformation into the *E. coli* strain B-95.ΔA[49]. Briefly, transformed bacteria were directly inoculated into a 2 L plastic baffled flask (Thomson Instrument Company) containing 200 mL optimised growth medium with 15 g/L tryptone, 30 g/L yeast extract, 8 mL/L glycerol (Promega), 10 g/L NaCl and shaken at 200 RPM overnight at 37 °C. High-density cultures were then reduced to room temperature and induced with 0.4 mM IPTG (Roche) for 6 h. Bacteria were harvested by centrifugation at 4000 g. The bacterial pellet was resuspended in a high salt buffer (1 M NaCl, 50 mM Imidazole, 50 mM monosodium phosphate, adjusted to pH 8.0) supplemented with complete EDTA-free protease inhibitors, 2 mM $MgCl_2$ and benzonase. Resuspended bacteria were lysed by homogenisation with an EmulsiFlex-C3 (Avestin) before centrifugation at 12,000 × g for 1 h and clarified through a 0.45-μm syringe filter to remove cellular debris. Protein was purified by immobilised metal affinity chromatography (IMAC) using Protino Ni-NTA agarose (Machery-Nagel). Captured protein was washed copiously with high salt buffer and a low salt buffer (100 mM NaCl, 50 mM Imidazole, 50 mM monosodium phosphate, adjusted to pH 8.0) before elution (300 mM NaCl, 450 mM Imidazole, 50 mM monosodium phosphate, adjusted to pH 8.0). Eluted mApple was concentrated and buffer exchanged into pH 7.4 PBS using 10 kDa

molecular weight cut-off Amicon centrifugal filters (Merck). Protein concentration was determined by A568 with e = 82,000 M$^{-1}$ cm$^{-1}$ [27].

## Labelling of PEI
PEI (Mn ~1200 g mol$^{-1}$, product #482595) at 1 mg mL$^{-1}$ in MilliQ water was incubated with 5 molar equivalents of Cyanine5 succinimidyl Ester (Lumiprobe) in a total volume of 35 μL for 2 h at room temperature. Removal of excess dye was achieved by 0.5 mL Zeba spin desalting columns (7 kDa molecular weight cut-off) which were equilibrated with PBS, according to manufacturer's instructions.

## Lentivirus production and transduction
HEK293-FT cells were seeded one day prior at 400,000 cells/well in 6-well culture plates. The following day, lipofectamine 3000 (ThermoFisher Scientific) was used to transfect the cells with transfer plasmid and third-generation lentiviral vectors to generate lentivirus. 48 h post transfection, HEK293-FT culture medium was clarified with a 0.45 μm springe filter before being applied to NIH-3T3 cells, seeded one day prior in a 12-well culture plate at 50,000 cells/well. NIH-3T3 cells were grown to ~80% confluency then selected with 2 μg mL$^{-1}$ puromycin for positive incorporation of the transfer gene.

## Transient expression of endosomal stage markers
NIH-3T3 cells expressing either TfR or TMEM106b fused mApple were seeded at 2500 cells/well in a black 96-well clear bottom plate. The following day, lipofectamine 3000 was used to transfect Rab5a-mEmerald or LAMP1-mEmerald plasmids. In both constructs the mEmerald resides on the cytosolic side of the endosomal membrane. Cells were imaged live 48 h later using LEICA SP8X FALCON Confocal system using a HC PL APO 86× 1.2NA water immersion objective. Excitation for mEmerald and mApple was from a SuperContinum WLL at 488 nm and 561 nm, respectively. Emission was collected to SMD HyD detectors at 500–560 nm for mEmerald and 580–695 nm for mApple with a pixel size of 133 nm.

## Fast fluorescence lifetime microscopy
Traditional time-correlated single-photon counting (TCSPC) is intrinsically slow, requiring long integration times. Here we used a Leica SP8 FALCON (FAst Lifetime CONtrast) microscope to acquire the FLIM data. The FALCON system uses pattern recognition analysis of digitised signal from the spectral single-photon counting detectors, and transforms this signal into photon arrival times. This approach allows for significantly higher photon flux, resulting in shorter integration times for each image[50]. mApple was excited at 561 nm with a repetition rate of 80 MHz and emission was detected from 571 to 660 nm. 8–16 lines were accumulated per capture to increase photon counts with a pixel size of 133 nm.

## mApple calibration
For the pH calibration, 5 μL of 75 μM mApple protein was combined with 120 μL of relevant pH buffer in a black 96-well clear bottom plate. For ionic strength calibration, 2x PBS (300 mM ionic strength) was used to create solutions of relevant ionic strengths by dilution with MilliQ water before combining mApple with these buffers in the ratio outlined above. The plate was then imaged on a LEICASP8X FALCON Confocal system as mentioned above. System was pre-warmed to 37 °C in the focal plane just above the surface of the plate. Fast fluorescence lifetimes (Fig. 1c) were calculated by applying a bin of 8 to the captured image then processing according to Eq. (2).

$$\text{Mean weighted lifetime} = \frac{\sum(\text{Pixel photon count} \times \text{Pixel fast flourescent lifetime})}{\sum \text{Photon counts in image}} \quad (2)$$

In the case of the G value calibration (Fig. 1g), a bin of 8 was applied to the captured image and the phasor G coordinates were averaged according to Eq. (3).

$$\text{Mean weighted G value (image)} = \frac{\sum(\text{Pixel photon count} \times \text{Pixel G value})}{\sum \text{Photon counts in image}} \quad (3)$$

G value calibration was fit with linear regression and 95% prediction bands were plotted in Prism. For the intensity calibration (Fig. 1f), intensity values from the acquired images were normalised to the maximum value at pH 7.4, then data was fitted to a four-parameter logistic sigmoidal fit in Prism. 95% prediction bands were plotted using prism. Uncertainty in the pH measurement for the intensity and G value calibrations were determined from the 95% asymmetric confidence interval.

## Live cell imaging
NIH-3T3 cells expressing the relevant fluorescent proteins were seeded one day prior in cell culture medium at 10,000 cells/well in a black 96-well clear bottom plates. Prior to imaging, cell media was replaced with pre-warmed (37 °C) imaging medium (Fluorobrite, 10% FBS) and incubated for at least 10 min before being inserted into the pre-warmed (37 °C, 5% CO$_2$) microscope. If BafA1 or PEI were to be added, they were diluted to a final concentration of 100 nM or 80 μg/mL, respectively, in imaging medium before replacing the original cell culture medium after the plate was inserted into the microscope. The same regions were imaged at relevant timepoints. In the case of Cy5-PEI treatment, cells were treated with Cy5-PEI at 80 μg/mL for the indicated time before stringent washing with imaging medium before imaging. It was not possible to image the same cells over the time course with Cy5-PEI treatment due to the high signal of Cy5-PEI in the surrounding medium. In the case of Cy5-PEI pDNA polyplex treatment, polyplexes were assembled by combining Cy5 labelled PEI (as above) and transfection grade PEI (PEI max linear, Polysciences, MW 40,000) at a weight ratio of 1:5, respectively, then adding EGFP encoding pDNA at a weight ratio of 1:40 (pDNA:PEI) in fluorobrite supplemented with 10% FBS to a final pDNA concentration of 2 μg/mL. This polyplex solution was incubated with NIH-3T3 cells transduced with TMEM-mApple for 4 or 6 h, after which the solution was removed and the cells were washed three times with fluorobrite. Cells were imaged after washing, then returned to the incubator for EGFP transfection assessment the following day.

## Transfection analysis of PEI/pDNA polyplexes
Twenty-four hours after the initial addition of PEI/pDNA polyplexes above, cells were detached from the imaging plate using TrypIE and the EGFP fluorescence was quantified by flow cytometry using a Stratedigm S1000EON with a 488 nm laser. Fluorescence emission was collected from 500 to 540 nm of ~10,000 events per sample. FCS3.0 files were exported using CellCapTure Analysis Software (Stratedigm, California, USA) and analysed using FlowJo (version 10, Becton, Dickinson and Company; 2021).

## Training of StarDist
The StartDist algorithm was trained using the ZeroCostDL4Mic[51] Google Colab notebook. The images for training were initially segmented using the pre-trained "versatile (fluorescent nuclei)" model using normalised images, percentile low = 0.5, percentile high = 99.8, probability threshold = 0.05 and overlap threshold = 0. This resulted in images with a large number of false positives, but with nearly all the endosomes identified. These images were then individually inspected and all the false positive ROIs were deleted. Eight 512 × 512 images with >250 endosomes identified per image were uploaded into the 2D

StarDist ZeroCostDL4Mic Google Colab notebook and training was performed using the default settings.

## Automated analysis of images

Images were exported by applying a preview filter with a value of 1000 and phasor threshold of 5. Exported images were analysed using custom FIJI (ImageJ) scripts. Briefly, to identify vesicles, the analyse_FLIM_Images_with_Stardist.ijm script employs a custom trained Stardist model (outlined above) which automatically detects endosome-like objects in the mApple intensity channel. The algorithm was run with the following settings: Normalize input = true, percentile low = 25, percentile high = 99.8, probability threshold = 0.5, overlap threshold = 0. The intensity channel of each image had an intensity threshold of 15 photons/pixel applied then was segmented into endosomes by the algorithm. The segmentation mask was then applied to the corresponding G value channel with an upper area cutoff of $2\,\mu m^2$. Intensity and G value data found within the detected endosome was then used to determine the average weighted G value of the vesicle according to Eq. (1). The mean weighted G value was used as it weights the mean value towards G values with higher photon representations in the vesicle. Note that intensity values correspond to the number of detected photons directly. Mean weighted G values were then converted to a mean weighted pH as per the linear trend observed in Fig. 1g. A custom R script was used to analyse and generate plots of this data.

To identify vesicles that were double positive for both mApple and PEI (Cy5), the Create_Double_Positive_mask.ijm script was used. Briefly, this script uses the same Stardist model outlined above to separately identify vesicles that are positive for mApple or PEI in their respective channels. These masks are then eroded by one pixel to limit the detection of vesicles that are close to each other, but not completely coincident. A mask for vesicles that have signal from both the mApple and PEI channels (using the original non-eroded mask for mApple) is then created, as well as a mask of vesicles that contain only mApple. The same script was also used to calculate the coincidence of mApple with Rab5a and LAMP1. The percentage coincidence number from each image was calculated by ratioing the number of double positive vesicles detected by the total number of mApple positive vesicles detected.

To determine the pH of vesicles that contain PEI vs those that do not contain PEI, the masks generated by the Create_Double_Positive_mask.ijm script were used in conjunction with the Analyse_FLIM_Images_with_precalculated_masks.ijm. This script works the same way as the analyse_FLIM_Images_with_Stardist.ijm script, except it used a predetermined mask instead of Stardist to identify the vesicles. To correlate the intensity of PEI in each vesicle with the pH of the vesicle, the Quantify_pH_mask&intensity.ijm script was used. This script takes the G value pH image, the PEI intensity image and double positive mask (generated from Create_Double_Positive_mask.ijm) and calculates the pH and total PEI (Cy5) intensity based on the supplied double positive mask.

## Statistics and reproducibility

No data were excluded from the analyses. The number of biological replicates and the statistical tests used to determine significance are outlined in the figure capture. No statistical method was used to pre-determine sample size.

## Reporting summary

Further information on research design is available in the Nature Research Reporting Summary linked to this article.

## Data availability

All data including microscopy images generated in this study are provided in the Source data file and are also available on figshare

(https://doi.org/10.6084/m9.figshare.20454867.v1). Source data are provided with this paper.

## Code availability

Custom ImageJ scripts used in this study are available with this manuscript as Supplementary Software files, and are also available on figshare (https://doi.org/10.6084/m9.figshare.20454867.v1).

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

## Acknowledgements

J.J.R. was supported by an Australian Government Research Training Program (RTP) Scholarship. A.P.R.J. was supported by an NHMRC Career Development Fellowship (GNT1141551) as well as ARC Discovery Projects (DP200100475, DP210103174) and NHMRC Ideas Grant (GNT2011963).

## Author contributions

J.J.R. performed all experimental work. A.P.R.J. developed the concept, A.P.R.J. and C.W.P. supervised the study. C.J.N. developed the ImageJ analysis script with input from A.P.R.J. and J.J.R. J.J.R. developed the R analysis script. A.P.R.J. and J.J.R. wrote the manuscript, C.J.N. and C.W.P. edited the manuscript.

## Competing interests

The authors declare no competing interests.
