## [Peer Review File · Nature Communications]

Reviewers' comments:

Reviewer #1 (Remarks to the Author):

The authors show that the fluorescent protein mApple exhibits a linear dependency of fluorescence lifetime vs pH. They show through various experiments that FLIM measurements allow to accurately determine the pH inside cells and in endosomal vesicles in particular. Measuring pH by FLIM rather than by (ratiometric) intensity measurements is of interest since it does not depend on the concentration of the probe. The linear dependence also has as an advantage that pH can be determined equally precise across its sensitivity range. The manuscript is well-written and easy to follow, except that figures and figure subpanels are not displayed in the same order as they are discussed in the text, which is somewhat confusing at times, and I would recommend to change this in a revised version.

My only remark is about the PEI experiments and the subsequent interpretation in terms of the absence of pH changes in the proton sponge mechanism. The authors show in one cell type (NIH-3T3) that solubilized PEI does not lead to measurable changes in vesicular pH after pinocytic uptake. They see this as evidence that buffering of pH may not be involved in endosomal escape by the proton sponge mechanism. However, uptake and intracellular trafficking by solubilized PEI may be quite different from actual PEI/DNA complexes. So, first of all, this experiment should be repeated with actual PEI/DNA complexes and not just solubilized PEI. PEI or DNA should be labeled so as to be able to tell in which endosomes complexes are actually present and perform pH analysis in those vesicles specifically. Furthermore, if the authors really want to make statements on the mechanism of endosomal escape by the proton sponge mechanism, they should in addition monitor endosomal escape events (see e.g. Rehman et al ACS Nano 7, 2013) and check if endosomes from which escape happens exhibited pH buffering or not. This would be needed to make strong conclusions about the involvement of pH buffering in the endosomal escape of PEI/DNA complexes. Preferably this would be done in several cell types.

Reviewer #2 (Remarks to the Author):

The manuscript from Rennick et al. uses the pH-sensitivity of the fluorescent protein (FP) mApple to explore variations in intracellular pH. To this end, the authors use a phasor-FLIM approach instead of relying on fluorescence intensity calculations. This is an elegant method, as FLIM is insensitive to fluorophore concentration. This convenient property allows to use mono FP-based sensors, therefore bypassing dual-FP probes and their potentially error-prone ratiometric intensity quantifications. To uncover the power of mApple in tracking pH-related events in cells, the authors focus on the roles of

TMEM106b and TfR in the endo/lysosomal degradation pathway. They do so by fusing each protein to mApple, and by following their FLIM behaviour on endo/lysosomal vesicles. The authors also couple phasor-FLIM to StarDist-based deep-learning approaches, to increase the number of vesicles analysed and with this, the statistical power of their analyses.

The manuscript is certainly very interesting, well written, and it builds on the power of FLIM to track intracellular events with superior spatiotemporal resolution.

I acknowledge that it might be difficult – if not impossible sometimes – to perform additional experiments during the pandemic. Nevertheless, I believe that the manuscript is far too preliminary to warrant acceptance in its present form.

The main points that require improvement are:

1. The authors should rephrase their manuscript when discussing about the suitability of their method to follow individual vesicles. What they provide throughout the manuscript are calculations on entire vesicle populations, and fail in discriminating between intracellular vesicles pools which could be of interest.
2. To reflect real changes in pH and how mApple-based constructs are capable of tracking them, the present findings should be backed up by KO/KD of early vs late autophagy genes and other chemicals (Chloroquine, Pepstatin...) to validate BafA1 finding. At present, the authors cannot exclude that the variations in mApple constructs are linked to a previously unreported side-effect of BafA1 on this particular FP, and that it is unrelated with the role of this inhibitor in blocking the autophagic flux.
3. Missing controls: a benchmarking analyses with at least (i) one pH-sensitive and one insensitive FP; (ii) an already-validated probe to follow pH variations (e.g. pHLare); (iii) LAMP1-mApple to visualize whether it is resistant into the lysosomal environment and its FLIM behaviour using phasor-FLIM. What would be the respective FLIM behaviours of these constructs? Is mApple really more convenient than the existing solutions?

Specific points are provided below:

MAJOR POINTS:

1. It would be important to provide evidence, at least in the in vitro system, on whether the pH sensitivity and linearity of mApple are reversible features. What would happen if pH was lowered and subsequently increased? What would be the time resolution of this process and would pHLIM be able to track it? This would provide important information for analyses in cells such as neurons or potentially in vivo, where pH changes are not always irreversible and might fluctuate over time.
2. Images are rather of bad quality throughout the manuscript, and it is particularly difficult to appreciate Red/Green colocalization levels. Please provide insets for all conditions where colocalization was shown, and quantify colocalization levels numerically.
3. When looking at colocalization images, how do the authors explain the observation that mApple constructs appear less vesicular in cells co-transfected with LAMP1-mEmerald compared to nearby, untransfected cells? Similarly, why mApple constructs aggregate in cells overexpressing Rab5A-

mEmerald? Could this be a transfection side-effect? The authors need to provide evidence that they are not over activating the lysosomal/endosomal pathways with their overexpression strategy, and that within these aggregates there are no pH-independent variations of lifetime and/or quenching events (which could affect lifetime calculations and which seem visible from the micrographs provided)

4. The authors should refine their quantification by isolating (i) cell surface-specific pixels of TfR and (ii) TfR-specific endo/lysosomal vesicles. Can the authors decorrelate the two populations using lifetime? That would be of relevance, as they seem to have two intrinsic lifetime values in the pictures, and the power of FLIM would be particularly suitable in identifying and following the fate of these two populations. Are these two separate pools never mixing between them, or is this approach so powerful that one can follow them in real time and infer on the functional relevance of each pool?

5. Building on the previous question, how do the authors reconcile the observation that TMEM106b-mApple vesicles, which should belong to the late endo/lysosomal pathway only, also display a broad variability in their pH spectrum? Is this expected and if not, does this indicate the presence of protein pools with functional relevance?

6. Could the authors explain why the StarDist training was made with 8 pictures only? This could potentially narrow down the capacity of the algorithm to recognize the experimental variability of the test images. Concerning the test images, could the authors explain why there is such a huge discrepancy between the number of TfR-mApple images (~30) and the TMEM-mApple ones (~150) used for the analysis? A 5X factor might be of relevance and if not, the authors should clearly explain why.

7. Concerning the results obtained with StarDist, it appears that the algorithm is detecting two classes of vesicles, and that one is discarded (the membrane-bound one). However, the data in Fig. 2 and 3 indicate that there is a quite significant pH gradient in the vesicles retained after DL. However, I wonder whether there is a pH range in the excluded vesicles as well, where the cutoff was made and whether there is a rationale in this exclusion (see point 4).

8. I do not totally agree with the authors on the interpretation of the BafA1 data in Fig. 4. These variations in pH they observe upon incubation with BafA1 could also be due to the fact that BafA1 blocks the autophagic flux. This induces the accumulation of autolysosomes, which are not being recycled. Could the overall FLIM spectrum they observe be a result of an increased overall vesicle number and if so, could the authors decorrelate vesicle numbers and lifetime variations? Again, the probe seems to aggregate in the perinuclear area. Can you discriminate whether these aggregates alter the overall pH calculations? These seem to have an intrinsic higher pH compared to the peripheral vesicles (which are fewer in number).

9. The authors use NIH-3T3 cells for all their analyses. However, would their method be compatible with subtle changes in pH within individual vesicles e.g. in specific cell types like neurons? How plastic or sufficiently sensitive to pH increase or decrease is your method (See point 1)?

10. In Fig. 5 again, I disagree with the interpretation of data as provided by the authors. What is quantified here is a global population, and given the modest uptake of Cy5-PEI it is not surprising that the results are not significantly different between treated and untreated cells. Are the uptake times of Cy5-PEI and unlabelled PEI comparable? Could you envisage concentrating your analysis on fluo-PEI (a blue/cyan-FP labelled PEI should avoid FRET effects between mApple constructs and PEI) to monitor

only the vesicles that are double-labelled? It could well be that the results are specific to a sub-pool of the vesicles, and are lost in a global quantification of vesicles such as the one performed in this Figure.

MINOR POINTS:

- Please rearrange Fig. 1 by presenting the panels in the order they are described in the corresponding results section. It is quite counterintuitive to go back and forth among panels to follow the story on this particular Figure.
- mEmerald has a pKa of 6.0, so it is likely not the best FP to follow acidic compartments as the lysosomes. Please compare mApple/mEmerald colocalization levels with those of fmApple used in conjunction with a pH-insensitive FP.
- I wonder why the authors show several fields of view to illustrate colocalization. Instead of multiplying the FOVs, please consider using insets. This would also allow to better appreciate the Merge panels.
- Fig S7: why is colocalization peaking at 120 min and then decreasing?
- What is the minimal number of photons used to calculate lifetime values for the vesicles? How can you normalize a sufficient photon budget with differences in pH? This is also a concern for saturated cells (see fig. 2B, saturated cell on the top). In this picture, lifetime seems higher. Does that imply that lifetime increases if fluorescence is saturated cells? How do the authors explain that?

Reviewer #3 (Remarks to the Author):

The authors in their manuscript "Resolving subcellular pH with a quantitative fluorescent lifetime biosensor" employed a lifetime-pH sensitive fluorescent protein, mApple, to quantify pH in cells. They showed how mApple lifetime is linearly depended on the pH, which differentiate this fluorescent protein to other pH-depended sensor.

They demonstrated the bio-sensing property of mApple by measuring pH in some cellular compartments. Moreover, they employed a deep-learning algorithm to detect intra-cellular vesicles and quantify, thus, the pH changes between vesicles and over time. The authors tested mApple pH-sensing properties by expressing mApple fused with two endo/lysosomal proteins, under two different drugs.

The manuscript is well-written and easy to read.

The performed experiments and the presented methodology are sound and well described. However, the reviewer is underwhelmed by the use of the deep learning algorithm in this context. Even if well-applied and established for detecting endo-lysosomal compartments, it seems applicable only to segment intra-cellular vesicles. This weakens the manuscript, as it does not demonstrate how the method could be suitable for multiple biological applications, and it's not directly relevant.

The reviewer believes that this fluorescent protein will broaden the application range for measuring physiological and pathological pH changes in specific cellular compartments; however, there are some additional comments/questions to be addressed:

1) The bio-sensing ability of mApple is tested here with pH ranging from 4.6 up to 7.4. However, some cellular compartments show also higher pH, such as pH 7.8-8. Why did the authors decide to test only this limited range of pH?

2) The lifetime calibration of mApple for different lifetimes (Fig1 c and d for the phasor representation) is performed, if correctly understood, for purified proteins in buffer solution. The same pH-lifetime calibration curve is used to interpret the pH changes in the cellular compartments. However, the cellular environment is more complex and various (confinement, different protein conformation, possible protein interaction can also change the lifetime values and these are not addressed in the calibration curve in vitro), thus, the reviewer is not convinced about the use of the same in vitro calibration curve also for the cell measurement.

The manuscript would be stronger if more controls measurement were performed, for example with ions of different valency or in cells, under different conditions or in different cellular compartments, with a limited range of physiological pHs.

4) The authors claimed that mApple could be used for several organelles and cellular structures, however only pH in endo/lysosomal vesicles was quantified. The manuscript would be stronger with other examples of cellular structures, as for example mitochondria. This would strengthen the relevancy of this method to biological applications.

5) What does fast-FLIM mean? What are the differences from conventional FLIM? Whereas this technique may be thoroughly explained in the previous papers cited in the manuscript, a more basic and detailed explanation would be appropriate.

6) The dark background for all the phasor plots presented complicates the interpretation of the data. The reviewer suggests a lighter background.

Dr. Perego Eleonora

Reviewer #4 (Remarks to the Author):

The manuscript by Rennick et al. describes a method to measure the luminal pH of organelles of the endocytic pathway. The authors describe the construction of chimeras of the pH-sensitive fluorescent protein mApple attached to the terminus of either transferrin receptors or TMEM106b that faces the organellar lumen. They proceeded to express these constructs and validate their subcellular distribution and used them to measure the pH by fluorescence lifetime imaging microscopy (FLIM). Lastly, Rennick et al applied this system to assess the effects of bafilomycin and of polyethyleneimine on pH.

The method is not particularly original: pH-sensitive fluorescent proteins (including mApple and superior variants) had been used many times in the past to measure intracellular pH; similar chimeric constructs had been described and employed in the last 15 years to target pH-sensitive and other probes to organelles; lastly FLIM had been used before effectively to measure ionic concentrations, including pH. The results obtained were, accordingly, predictable and replicate earlier results from several other laboratories. The pH of the early and recycling endosomes reported here is consistent with multiple earlier determinations, as is the pH reported for the late endosomal/lysosomal compartment. Similar values have been reported in original papers, reviews and even textbooks.

The reported effects of bafilomycin are equally well established and were predictable. Lastly, the putative effects of polyethyleneimine on pH had been disproven before and the (negative) findings here are, again, confirmatory.

In summary, this manuscript is essentially the combination of FLIM with targeting of pH-sensitive chimeric proteins (which are in fact somewhat mistargeted as a result of overexpression), techniques that were established and validated earlier. The editor must decide at his/her discretion whether this contribution warrants publication in Nature Communications.

Reviewer 1

We thank the reviewer for highlighting that “*measuring pH by FLIM rather than by (ratiometric) intensity measurements is of interest since it does not depend on the concentration of the probe*” and noting our technique has the “*advantage that pH can be determined equally precise across its sensitivity range*”.

They raise a couple of points regarding the PEI and proton sponge aspects of the paper. We have performed additional experiments to address these points as outlined below.

1) Uptake and intracellular trafficking by solubilized PEI may be quite different from actual PEI/DNA complexes. So, first of all, this experiment should be repeated with actual PEI/DNA complexes and not just solubilized PEI. PEI or DNA should be labeled so as to be able to tell in which endosomes complexes are actually present and perform pH analysis in those vesicles specifically.

We thank the reviewer for identifying that PEI/DNA complexes may buffer differently from solubilised PEI. To address this, we have performed additional experiments to investigate the pH buffering of PEI/DNA complexes. We have updated the manuscript as follows:

“To further investigate the potential buffering effect of PEI, we probed the pH of endosomal compartments containing PEI/DNA complexes. pDNA that encodes for EGFP was complexed with Cy5 labelled PEI. The polyplexes were incubated with TMEM106b-mApple transduced NIH-3T3 cells (2µg/mL DNA concentration) for 4 or 6 hours, and both incubation times resulted in strong GFP in ~50% of cells after 24 hours (Fig. S18). Using the StarDist algorithm, we identified all the mApple positive vesicles as well a Cy5 positive vesicles that contain the PEI/DNA polyplexes. We then measured the pH of the double Cy5/mApple positive endosomes and compared them to the pH of the Cy5 negative, mApple positive endosomes (Fig. S19). Confirming the result observed for PEI by itself, the pH of endosomes containing PEI/DNA polyplexes was not significantly different to the pH of PEI/DNA negative endosomes (both ~pH 5.5 – Fig. S20).”

2) Furthermore, if the authors really want to make statements on the mechanism of endosomal escape by the proton sponge mechanism, they should in addition monitor endosomal escape events (see e.g. Rehman et al ACS Nano 7, 2013) and check if endosomes from which escape happens exhibited pH buffering or not. This would be needed to make strong conclusions about the involvement of pH buffering in the endosomal escape of PEI/DNA complexes.

We thank the reviewer for suggesting the idea of directly imaging of endosomal escape, however, we disagree that this will give further insight into the proposed proton sponge mechanism. This is for 2 reasons.

- 1) Our method measures the pH of individual endosomes (rather than an average of all the endosomes). This means that if there was a population of endosomes with buffered pH it would be present in our pH histograms. This was not observed. To clarify this point, we have revised the manuscript as follows.

“Furthermore, by using Cy5 labelled PEI/DNA complexes, we were able to measure the pH of individual PEI/DNA positive endosomes, and compare the pH to endosomes in the same cell without PEI/DNA. There was no difference in the pH of TMEM106b-mApple vesicles with or without PEI.”

- 2) Directly imaging endosomal escape as outlined in the ACS Nano paper has some potential limitations. It has been shown that exposing fluorescently labelled materials to light, even for a short period of time, can induce endosomal escape¹. Therefore, it is difficult to be sure that sudden bursts of endosomal escape observed using live cell microscopy are due to PEI induced escape, or the photo-induced escape. Therefore we do not think these additional experiments will add to our conclusions.

As outlined in our response to the first point above, to demonstrate that endosomal escape has occurred, we used a GFP plasmid in our new PEI/DNA experiments and measured GFP expression in the cells 48 hours after transfection. These results showed that ~50% of cells expressed GFP, indicating that the PEI transfection was successful and endosomal escape had occurred.

Reviewer 2

We thank the reviewer for highlighting our work describes “*an elegant method*” that “*is certainly very interesting, [and] well written*”.

We also thank the reviewer for making several useful suggestions for additional experiments and alternative interpretations of our results. We have performed these additional experiments and included discussions to clarify all the findings in our work. A detailed response to all the points they raised is outlined below.

1. The authors should rephrase their manuscript when discussing about the suitability of their method to follow individual vesicles. What they provide throughout the manuscript are calculations on entire vesicle populations, and fail in discriminating between intracellular vesicles pools which could be of interest.

We think the reviewer has misunderstood our method and the analysis that it gives. Our technique does indeed allow us to follow the pH of individual vesicles. The pH histograms shown in figures 3, 4 and 5 are the distribution of pH measured from individual vesicles. To make this point clearer, we have revised the manuscript as follows.

“The histograms show the pH of individual vesicles and highlight the variation in pH between individual vesicles in the cell. The ability to observe the distribution of pH within the cell is a key advantage of our pHlim technique.”

2. To reflect real changes in pH and how mApple-based constructs are capable of tracking them, the present findings should be backed up by KO/KD of early vs late autophagy genes and other chemicals (Chloroquine, Pepstatin...) to validate BafA1 finding. At present, the authors cannot exclude that the variations in mApple constructs are linked to a previously unreported side-effect of BafA1 on this particular FP, and that it is unrelated with the role of this inhibitor in blocking the autophagic flux.

We thank the reviewer for pointing out that BafA may affect the lifetime of mApple. To address this, we have performed additional experiments which show BafA has no effect on the lifetime of mApple, and revised the manuscript as follows.

“We first confirmed that the presence of bafA1 did not influence the lifetime of mApple (Fig. S13).”

We have also performed an additional experiment to show that the addition of chloroquine also induces a rapid change in endosomal pH. We have amended the manuscript as follows to reflect this point.

“The addition of chloroquine also resulted in a similar increase in endosomal pH (Fig. S14, 15).”

We don't believe that KO/KD studies of early and late autophagy genes will add to the conclusions we are making in this paper. Our conclusions from this section of work are that:

- 1) mApple can be used to dynamically assess the pH of subcellular compartments
- 2) The addition of BafA increases the pH of the endosomal compartments

Neither of these conclusions are dependent on the presence of early or late autophagy genes.

3. Missing controls: a benchmarking analyses with at least (i) one pH-sensitive and one insensitive FP; (ii) an already-validated probe to follow pH variations (e.g. pHLare); (iii) LAMP1-mApple to visualize whether it is resistant into the lysosomal environment and its FLIM behaviour using phasor-FLIM. What would be the respective FLIM behaviours of these constructs? Is mApple really more convenient than the existing solutions?

In the original version of this manuscript we included the control of pH dependant intensity of mApple to demonstrate the advantage of FLIM measurements over intensity measurements. However, to expand the number of controls, we have included the intensity and lifetime data for muGFP (a pH sensitive protein), pHlourin (a validated pH probe) and mCherry (a pH insensitive protein). We have revised the manuscript as follows to reflect these additional experiments:

“The linear dependence of mApple lifetime (G value) across the physiological pH range is in contrast to other commonly employed fluorescent proteins. muGFP²⁹ shows a similar drop in fluorescence intensity from pH 7.4 to 4.6 (~90%), however the G value remains consistent across this pH range (Fig. S2a). pHlourin¹⁸, a fluorescent protein developed specifically as a pH sensor exhibits a linear drop in intensity from pH 7.6 to 6, but is not sensitive to pH below this range. The G value of pHlourin shows some dependence on pH between 7.6 to 6 (decreasing from G = 0.55 at pH 7.6 to G = 0.45 at pH 6), however magnitude of the lifetime change and the range over which the change occurs is less than observed for mApple (Fig. S2b). The fluorescence intensity of mCherry³⁰ is largely insensitive to pH (with a 20% drop in intensity below pH 5.5) and has no change in the G value across the physiological range (Fig. S2c). Each of these points highlight the advantages of using mApple as a pH biosensor.”

The majority of LAMP1 fusion proteins have the fluorescent protein fused to the C-terminus, resulting in the fluorescent protein residing on the cytosolic side of the lysosomes. We attempted to generate an N-terminal mApple LAMP1 fusion protein, which would result in a luminal fluorophore, however rather than localising to lysosomes, we observed fluorescent signal throughout the cell.

1. It would be important to provide evidence, at least in the in vitro system, on whether the pH sensitivity and linearity of mApple are reversible features. What would happen if pH was lowered and subsequently increased? What would be the time resolution of this process and would pHLIM be able to track it? This would provide important information for analyses in cells such as neurons or potentially in vivo, where pH changes are not always irreversible and might fluctuate over time.

We thank the reviewer for raising this important point. We have performed an additional experiment to demonstrate the reversible nature of the pH response and revised the manuscript as follows:

“Furthermore, the pH induced lifetime change is reversible (Fig. S4), enabling dynamic changes in pH to be measured.”

2. Images are rather of bad quality throughout the manuscript, and it is particularly difficult to appreciate Red/Green colocalization levels. Please provide insets for all conditions where colocalization was shown, and quantify colocalization levels numerically.

We apologise for the use of Red/Green colocalization, which can be difficult to observe for colour-blind readers. We have changed the colours used in the supporting information to include colour-blind friendly colour schemes.

Numerically calculating the colocalization using Pearson’s or Mander’s coefficients is not appropriate for these images. The LAMP1 and Rab5a signal is on the cytosolic domain of the vesicles and the mApple signal is inside. While some colocalization can be observed at the edge of the vesicles, the centre of the vesicle does not register as ‘colocalized’. To overcome this, we used the StarDist algorithm to identify all the mApple and mEmerald positive vesicles, and used a binary overlap morphological filter to determine the percent of mApple endosomes that double positive for mEmerald. These values are now included in Figure S7 in the supplementary information.

3. When looking at colocalization images, how do the authors explain the observation that mApple constructs appear less vesicular in cells co-transfected with LAMP1-mEmerald compared to nearby, untransfected cells? Similarly, why mApple constructs aggregate in cells overexpressing Rab5A-mEmerald? Could this be a transfection side-effect? The authors need to provide evidence that they are not over activating the lysosomal/endosomal pathways with their overexpression strategy, and that within these aggregates there are no pH-independent variations of lifetime and/or quenching events (which could affect lifetime calculations and which seem visible from the micrographs provided)

We disagree with the reviewer’s interpretation of these images.

We cannot observe any difference between the vesicular nature of the cells that are co-transfected with LAMP1-mEmerald/mApple and the mApple only cells. The only example we could observe of this phenomena is highlighted in the image below. The cell highlighted with the arrow is flatter than the cells around it and thus signal is observed on the surface of the cell (as noted in the manuscript TfR is present on the surface and inside the cell).

We have reviewed all the images of Rab5A-mEmerald/mApple expressing cells and do not observe a different pattern for mApple in the Rab5A-mEmerald expressing cells.

4. *The authors should refine their quantification by isolating (i) cell surface-specific pixels of TfR and (ii) TfR-specific endo/lysosomal vesicles. Can the authors decorrelate the two populations using lifetime? That would be of relevance, as they seem to have two intrinsic lifetime values in the pictures, and the power of FLIM would be particularly suitable in identifying and following the fate of these two populations. Are these two separate pools never mixing between them, or is this approach so powerful that one can follow them in real time and infer on the functional relevance of each pool?*

We thank the reviewer for pointing out this additional benefit of the pHlim technique. We have performed this analysis on the TfR-mApple images and have shown that the surface TfR and internalised TfR can be easily distinguished using the lifetime measurements.

“These different populations of TfR-mApple (high pH surface and early endosome, and low pH late endosome/lysosome) can be easily distinguished by the lifetime measurements (Fig. S8).”

5. *Building on the previous question, how do the authors reconcile the observation that TMEM106b-mApple vesicles, which should belong to the late endo/lysosomal pathway only, also display a broad variability in their pH spectrum? Is this expected and if not, does this indicate the presence of protein pools with functional relevance?*

We disagree that the TMEM106b vesicles show broad variability in their pH. In comparison to the TfR vesicles, the TMEM106b vesicles show a narrower pH distribution. From Figures 3 and S10 the average full width half maximum (FWHM) of TMEM106b-mApple vesicles is ~1 pH unit, whereas TfR-mApple vesicles have a FWHM of >2 pH units. Some degree of variation is to be expected as we do not expect all vesicles to have the same pH.

6. *Could the authors explain why the StarDist training was made with 8 pictures only? This could potentially narrow down the capacity of the algorithm to recognize the experimental variability of the test images. Concerning the test images, could the authors explain why there is such a huge discrepancy between the number of TfR-mApple images (~30) and the TMEM-mApple ones (~150) used for the analysis? A 5X factor might be of relevance and if not, the authors should clearly explain why.*

While only 8 images were required for STARDIST training, each image had >100 individually segmented endosomes, with a total of >1000 individual endosomes identified. With this data, the algorithm rapidly converged to accurately identify the endosomal compartments. The accuracy and robustness of the algorithm is shown in figure S21.

The Stardist analysis was performed on ~30 TfR-mApple cells (not images) and ~150 TMEM-mApple cells (not images). The TfR-mApple cells contain a higher number of endosomes per cell, therefore a lower number of images was required to get robust statistical analysis. The higher number of TfR vesicles per cell than TMEM vesicles per cells was expected, as multiple early endosomes converge into late endosomes/lysosomes – thus there are less late/endosomes than early endosomes in the cell. We were aware that there were fewer TMEM vesicles per cell so acquired additional images to compensate for this. For completeness we included the analysis of all the cells we acquired for TMEM, rather than stopping the analysis when we reached a similar number of vesicles to the TfR cells. The additional data does not affect the conclusions or analysis of the data.

7. Concerning the results obtained with StarDist, it appears that the algorithm is detecting two classes of vesicles, and that one is discarded (the membrane-bound one). However, the data in Fig. 2 and 3 indicate that there is a quite significant pH gradient in the vesicles retained after DL. However, I wonder whether there is a pH range in the excluded vesicles as well, where the cutoff was made and whether there is a rationale in this exclusion (see point 4).

The StarDist algorithm is not detecting different classes of vesicle. It is designed to detect all vesicle structures, and all detected vesicles are included in the analysis. No vesicles detected by the StarDist algorithm are discarded. The StarDist algorithm is deliberately employed to eliminate surface bound mApple, but this by its definition is not in a vesicle. The reviewer is correct that Fig 2 and 3 indicate a pH gradient in the vesicles. This is consistent with the pH histogram shown in Figure 3e, which is indicative of the established acidification processes of early endosomes.

8. I do not totally agree with the authors on the interpretation of the BafA1 data in Fig. 4. These variations in pH they observe upon incubation with BafA1 could also be due to the fact that BafA1 blocks the autophagic flux. This induces the accumulation of autolysosomes, which are not being recycled. Could the overall FLIM spectrum they observe be a result of an increased overall vesicle number and if so, could the authors decorrelate vesicle numbers and lifetime variations? Again, the probe seems to aggregate in the perinuclear area. Can you discriminate whether these aggregates alter the overall pH calculations? These seem to have an intrinsic higher pH compared to the peripheral vesicles (which are fewer in number).

The reviewer proposes an interesting hypothesis that BafA1 works by blocking autophagic flux. As outlined below, the pHlim results do not back up this hypothesis, but this highlights the power of the pHlim technique and how it can give insight into biological mechanisms that would not be possible with existing methods. We have revised the manuscript as follows to highlight this point.

“BafA1 is proposed to increase endosomal pH by inhibiting both the V-ATPase H⁺ pump³⁵, which prevents acidification of vesicles, and the SERCA Ca²⁺ pump³⁹, which disrupts autophagosome/lysosomal fusion. To investigate which mechanisms plays the primary role in increasing the pH of TMEM+ vesicles we analysed the distribution of vesicle pH and the number of vesicles throughout the BafA1 treatment. If V-ATPase H⁺ pump inhibition is the primary mechanism for increasing the endosomal pH, we would expect to see the number of vesicles remain constant and for a steady increase in the pH of all the endocytic vesicles. However, if inhibition of the SERCA Ca²⁺ pump is the primary mechanism, we would expect to see the number of vesicles increase due to the inhibition of autophagosome/lysosomal fusion. We would also expect to observe two vesicle populations; the initial population of vesicles

which would exhibit a lower pH; and a new population of vesicles with a higher pH that are unable to fuse to the autophagosome/lysosomal compartments. Conventional analysis of the average pH inside the cell would not be able to distinguish between these two mechanisms, as in both cases the overall pH of the cell would increase. However, by analysing the individual vesicles we observed a single pH distribution with increasing (and narrower) pH, with a similar number of vesicles present throughout the experiment (Fig. S22). The lack of a second population of vesicles with a higher pH and the consistent number of vesicles suggests that over the 60 min time course of the experiment, V-ATPase H⁺ pump inhibition plays the major role in disrupting the endosomal pH.”

Accumulation of TMEM in the perinuclear region is expected, as the majority of endo/lysosomes are located in this region². This is not an indication of aggregation.

9. The authors use NIH-3T3 cells for all their analyses. However, would their method be compatible with subtle changes in pH within individual vesicles e.g. in specific cell types like neurons? How plastic or sufficiently sensitive to pH increase or decrease is your method (See point 1)?

The method could be used to track the pH within any cell that can be transfected with the mApple fusion. To demonstrate this we have performed the experiments in HEK293 and revised the paper as follows.

“Similar results were observed in HEK293 cells (Fig. S11,12).”

10. In Fig. 5 again, I disagree with the interpretation of data as provided by the authors. What is quantified here is a global population, and given the modest uptake of Cy5-PEI it is not surprising that the results are not significantly different between treated and untreated cells. Are the uptake times of Cy5-PEI and unlabelled PEI comparable? Could you envisage concentrating your analysis on fluo-PEI (a blue/cyan-FP labelled PEI should avoid FRET effects between mApple constructs and PEI) to monitor only the vesicles that are double-labelled? It could well be that the results are specific to a sub-pool of the vesicles, and are lost in a global quantification of vesicles such as the one performed in this Figure.

As noted previously, we think they have misunderstood the nature of the pH analysis. While the histograms represent the global population of endosomes, they show the pH of each individual vesicle. If there was a separate population (even a very small number of vesicles) with a lower pH, this would be observed in the histogram. Also, while there is limited signal from PEI at early time points, at the 4 and 6 hour time points, ~50% of the vesicles contain significant PEI signal. Therefore, we would expect that if there was an alteration of the endosomal pH it would be observed.

As the reviewer suggested, we performed additional experiments with Cy5 labelled PEI/DNA complexes and measured the pH of the double positive vesicles and compared the pH to the mApple only vesicles (Fig. S18, S19, S20). We did not observe any difference in the two populations of vesicles. The details of these additional experiments are outlined in our response to reviewer 1.

MINOR

POINTS:

- Please rearrange Fig. 1 by presenting the panels in the order they are described in the corresponding results section. It is quite counterintuitive to go back and forth among panels to follow the story on this particular Figure.

We have updated the order of panels in this figure.

- mEmerald has a pKa of 6.0, so it is likely not the best FP to follow acidic compartments as the lysosomes. Please compare mApple/mEmerald colocalization levels with those of fmApple used in conjunction with a pH-insensitive FP.

The mEmerald is fused to the C-terminus of LAMP, which means the mEmerald is located on the cytosolic side of the endosomes. This means it is unaffected by the pH of the lysosome. We have updated the experimental details to clarify this point.

“In both constructs the mEmerald resides on the cytosolic side of the endosomal membrane.”

- I wonder why the authors show several fields of view to illustrate colocalization. Instead of multiplying the FOVs, please consider using insets. This would also allow to better appreciate the Merge panels.

We chose to show multiple fields of view to demonstrate that the cells we are showing are representative of the whole sample. We have included the original images in the supplementary data files to allow full investigation of the data.

- Fig S7: why is colocalization peaking at 120 min and then decreasing?

We disagree that the colocalization peaks at 120 min. To make this clearer, we have included Supplementary figure 17, which shows the coincidence analysis of the PEI with mApple over the time course of the experiment.

- What is the minimal number of photons used to calculate lifetime values for the vesicles? How can you normalize a sufficient photon budget with differences in pH? This is also a concern for saturated cells (see fig. 2B, saturated cell on the top). In this picture, lifetime seems higher. Does that imply that lifetime increases if fluorescence is saturated cells? How do the authors explain that?

As outlined in the experimental section, the minimum number of photons used to calculate the lifetime is 15 photons per pixel. Saturation of signal in FLIM imaging is different from conventional confocal or widefield microscopy. Saturation for FLIM occurs if more than one photon reaches the detector during each laser pulse. If two photons reach the detector during the same pulse, then the data is excluded. The images are visualised as the sum of all photons detected at that individual pixel. For images such as the one highlighted by the reviewer in Figure 2B, the signal used to calculate the pH is not saturated. For the cell the reviewer refers to, the confocal slice is capturing the surface of the cell, which corresponds to the high signal and higher pH.

Reviewer 3

We thank reviewer 3 for noting that “*the manuscript is well-written and easy to read*” and “*the performed experiments and the presented methodology are sound and well described.*”

However, the reviewer is underwhelmed by the use of the deep learning algorithm in this contest. Even if well-applied and established for detecting endo-lysosomal compartments, it seems applicable only to segment intra-cellular vesicles. This weakens the manuscript, as it does not demonstrate how the method could be suitable for multiple biological applications, and it's not directly relevant.

We disagree with the reviewer's opinion that quantifying the pH of intra-cellular vesicles limits the scope and application of this work. Detecting the pH of intra-cellular vesicles is more useful than measuring the average pH of the whole cell. pH gradients across subcellular compartments play a crucial role in many cell functions, including membrane transport, energy production and degradation pathways. Our method enables the pH of these compartments to be quantified. Measuring the average pH of the cell does not enable changes in the distribution of vesicle pH. For example, as highlighted in response to point 8 from reviewer 2, by measuring the distribution of vesicle pH, we could distinguish between the two proposed mechanisms by which BafA increases endosomal pH. This would not be possible with existing techniques that only measure the average cell pH.

The work presented in this paper focuses on the pH of endosomal vesicles, but it could be easily applied to measure any sub-cellular compartment. AI approaches have already been demonstrated by others to identify mitochondria³, nucleus⁴ and could be developed for any other distinct cellular compartment.

We also note that the mApple pH sensor can be used without the AI algorithm (as shown in Figure 2). If used this way, the mApple sensor is still a significant advance over other pH sensors due to the linear behaviour of mApple lifetime (resulting in lower uncertainty in the inferred pH) and there is no need for a second reference fluorophore.

1) The bio-sensing ability of mApple is tested here with pH ranging from 4.6 up to 7.4. However, some cellular compartments show also higher pH, such as pH 7.8-8. Why did the authors decide to test only this limited range of pH?

We calibrated the mApple sensor from 4.5 to 7.4 as this was the range of pH values we observed for the 3 mApple constructs we used (cytosol, TfR and TMEM).

2) The lifetime calibration of mApple for different lifetimes (Fig 1 c and d for the phasor representation) is performed, if correctly understood, for purified proteins in buffer solution. The same pH-lifetime calibration curve is used to interpret the pH changes in the cellular compartments. However, the cellular environment is more complex and various (confinement, different protein conformation, possible protein interaction can also change the lifetime values and these are not addressed in the calibration curve in vitro), thus, the reviewer is not convinced about the use of the same in vitro calibration curve also for the cell measurement.

and

3) The manuscript would be stronger if more controls measurement were performed, for example with ions of different valency or in cells, under different conditions or in different cellular compartments, with a limited range of physiological pHs.

We thank the reviewer for this suggestion. The reviewer is correct that the pH calibration for mApple shown in Figure 1 was performed in buffer solutions. We demonstrated that the lifetime is not dependant on protein or salt concentration, however the reviewer is correct that the intracellular environment is more complex. To address this point we used digitonin to permeabilise cells expressing TfR-mApple and added different buffer solutions to alter the pH. These results are shown in Figure S4a.

The pH measured in the permeabilised cells has close to a linear relationship to the expected pH. The slight deviation from linearity is likely due to incomplete permeabilization of the cells and difficulty equilibrating the pH throughout the entire cell. Therefore, we have continued to use the free protein calibration because it is challenging to ensure the cells are completely permeabilized and the pH is equilibrated throughout the entire cell. The close relationship of the permeabilised cell calibration to the standard buffer calibration indicates that the more complex biological environment does not affect the lifetime of mApple.

4) The authors claimed that mApple could be used for several organelles and cellular structures, however only pH in endo/lysosomal vesicles was quantified. The manuscript would be stronger with other examples of cellular structures, as for example mitochondria. This would strengthen the relevancy of this method to biological applications.

In this work we demonstrated the ability to measure cytosolic pH and endo/lysosomal vesical pH. In the revised manuscript (at the suggestion of reviewer 2), we have expanded this analysis to include segmenting the cell membrane based on the difference in pH of endocytosed TfR and surface bound TfR. The focus of this manuscript was to demonstrate the ability to quantify subcellular pH and demonstrate relevant biological phenomena (BafA1 induced pH change and PEI proton sponge effect). While it would be possible to measure the pH of other organelles such as the mitochondria and demonstrate the effects of mitochondrial dysfunction on pH, we feel this is beyond the scope of this manuscript.

5) What does fast-FLIM mean? What are the differences from conventional FLIM? Whereas this technique may be thoroughly explained in the previous papers cited in the manuscript, a more basic and detailed explanation would be appropriate.

We thank the reviewer for highlighting that we didn't make this point clear. We have included the following section in the experimental section to explain the details of fast-FLIM.

“Traditional time-correlated single photon counting (TCSPC) is intrinsically slow, requiring long integration times. Here we used a Leica SP8 FALCON (FAst Lifetime CONtrast) microscope to acquire the FLIM data. The FALCON system uses pattern recognition analysis of digitised signal from the spectral single-photon counting detectors, and transforms this signal into photon arrival times. This approach allows for significantly higher photon flux, resulting in shorter integration times for each image. mApple was excited at 561 nm with a

repetition rate of 80MHz and emission was detected from 571-660nm. 8-16 lines were accumulated per capture to increase photon counts with a pixel size of 133 nm.”

6) The dark background for all the phasor plots presented complicates the interpretation of the data. The reviewer suggest a lighter background.

Unfortunately the output from the software used to generate the phasor plots automatically generates plots with a black background, and it is not possible to alter this.

Reviewer 4

Reviewer 4 raised a number of points around the originality of the work in the paper. We think these comments may reflect an incomplete understanding of the current literature and misinterpretation of the results presented in this paper.

The method is not particularly original: pH-sensitive fluorescent proteins (including mApple and superior variants) had been used many times in the past to measure intracellular pH;

As summarised in the introduction to our paper, the reviewer is correct that pH sensitive proteins have previously been used to measure cellular pH. However, as highlighted in the introduction there are a number of limitations with these studies. Principally, most assays rely on intensity measurements which:

- a. Require a second fluorophore to decouple concentration from pH
- b. Have low signal to noise and are significantly less accurate than our pHLIM measurements

The poor of accuracy of intensity based measurements (>1 pH unit), means they are not suitable for detecting subtle yet critical pH changes. Our pHLIM technique addresses this limitation.

similar chimeric constructs had been described and employed in the last 15 years to target pH-sensitive and other probes to organelles;

While pH sensitive probes have been used to probe the pH of specific organelles, these studies have not quantified the pH of individual vesicles and the distribution of pH within the cell. These studies have been limited to measuring the average pH inside the cell. The significant advantage of our technique is highlighted by our ability to distinguish between the proposed mechanisms for BafA1 action (as highlighted in our response to reviewer 2 above). The previous chimeric studies have also been limited by the sensitivity issues highlighted above.

lastly FLIM had been used before effectively to measure ionic concentrations, including pH.

While FLIM measurements have been used to measure pH, the previous studies:

- a) Used fluorophores with non-linear lifetime to pH response, which limits the pH range where they can accurately interpret pH, and significantly complicates the analysis.
- b) Have not quantified the distribution of pH within individual vesicles in the cell

Our pHLIM technique exhibits a linear response to pH with <0.1 pH uncertainty across the physiologically relevant pH range. Furthermore, we have quantified the distribution of pH

within individual cells, enabling us to identify different populations of vesicles and changes that occur in them.

The results obtained were, accordingly, predictable and replicate earlier results from several other laboratories. The pH of the early and recycling endosomes reported here is consistent with multiple earlier determinations, as is the pH reported for the late endosomal/lysosomal compartment. Similar values have been reported in original papers, reviews and even textbooks.

We agree the pH measured using our technique agrees with the well established literature in this space. The novelty of the paper does not lie in validating the pH of the different subcellular compartments. The novelty lies in:

- 1) Being able to more accurately measure the sub-cellular pH (to a point where subtle changes can be measured that previously wasn't possible)
- 2) Being able to measure the distribution of pH within the cell, rather than just measuring the average pH of the cell

The reported effects of bafilomycin are equally well established and were predictable.

As highlighted by the comments from reviewer 2, the effects of BafA1 were not well established or predictable. Our results here shed new light on the mechanism of endosomal pH changes induced by BafA1, namely that the primary mechanism of pH increase is inhibition of V-ATPase H⁺ pumps, and not inhibition of vesicle fusion.

Lastly, the putative effects of polyethyleneimine on pH had been disproven before and the (negative) findings here are, again, confirmatory.

While there have been a number of number of reports seeking to disprove the proton sponge effect, the proton sponge effect is still widely debated in the literature⁵. Part of the reason for the debate lies in the lack of direct evidence for measuring the subcellular pH of individual endosomes. Our results here provide this direct evidence.

References:

1. Srinivasan, D. *et al.* Conjugation to the Cell-Penetrating Peptide TAT Potentiates the Photodynamic Effect of Carboxytetramethylrhodamine. *PLoS One* **6**, e17732 (2011).
2. Cabukusta, B. & Neefjes, J. Mechanisms of lysosomal positioning and movement. *Traffic* **19**, 761–769 (2018).
3. Fogo, G. M. *et al.* Machine learning-based classification of mitochondrial morphology in primary neurons and brain. *Sci. Rep.* **11**, 5133 (2021).
4. Schmidt, U., Weigert, M., Broaddus, C. & Myers, G. Cell Detection with Star-Convex Polygons. in *Lecture Notes in Computer Science (including subseries Lecture Notes in Artificial Intelligence and Lecture Notes in Bioinformatics)* vol. 11071 LNCS 265–273 (2018).
5. Vermeulen, L. M. P., De Smedt, S. C., Remaut, K. & Braeckmans, K. The proton sponge hypothesis: Fable or fact? *Eur. J. Pharm. Biopharm.* **129**, 184–190 (2018).

REVIEWER COMMENTS

Reviewer #1 (Remarks to the Author):

The authors performed part of the suggested experiments to draw more reliable conclusions regarding the proton sponge effect. In the revised manuscript they looked at the pH of endosomes containing actual pDNA/PEI complexes and on the population level found no difference with the pH in endosomes not containing complexes. It is, however, important to realize that the analysis was done on a population level, i.e. averages were calculated which were not significantly different. This is where the problem lies regarding the conclusions that are drawn. Endosomal escape by PEI complexes is known to be highly inefficient in the sense that at best a few percentage of endocytosed complexes actually manage to escape. This has been attributed to variability in PEI content per endosome and the fact that endosomes can have different sizes. Indeed, to induce a sufficient amount of osmotic pressure through the proton sponge mechanism, there obviously should be a sufficient amount of polymer present in a particular endosome. Apparently this condition is met in only a few percentage of cases (i.e. polyplex containing endosomes). Therefore, it is reasonable to expect that the buffering action will only be present in a few percent of endosomes, and this may be obscured when analysis is done on averages. This is also exactly why I had proposed to look at endosomal escape events, because only then you can draw conclusions whether buffering is related to the escape events. The authors decided not to do this, and I will not insist on it since it's not the major point of the paper. Nevertheless I do insist on a more thoughtful discussion, including the above mentioned limitations of the performed analysis, before drawing the in my opinion oversimplified conclusion that buffering is not part of the proton sponge effect.

Reviewer #2 (Remarks to the Author):

First of all, I'd like to sincerely congratulate the authors for the impressive experimental work done during the revision phase. The manuscript has greatly improved, and the characterization of the properties of mApple in response to pH is now more careful. The conclusions they reach are more supported by data, resulting in an overall amelioration of the manuscript. I would also like to thank them for taking the time to explain their reasoning, both to me and in the manuscript. I feel that the reading of the manuscript is now easier for the general audience and to me, as a potential end-user of their tool.

I have very few comments, which can be found here below. However, the explanations provided in the response file reinforced the impression that I don't see a clear advantage of this tool and approaches presented in the manuscript, compared to the ones already existing in the literature.

1. As far as I understand, Fig S13 reports on data obtained with a purified mApple protein. If that is the case, please consider adding that it is the purified protein we're looking at, and not at cells expressing a vector coding for mApple alone.

2. Point 8: the effect of BafA1 in blocking the autophagy flux is not my hypothesis, but it has been experimentally verified. Please refer to <https://doi.org/10.1080/15548627.2020.1797280> for further details. In this light, the effect of BafA1 the authors observe on the overall vesicle number is very surprising and against what previously shown, and I wonder whether this is linked to a specific feature of the cells they use. In this light, the KO/KD of key genes involved in the autophagosomal pathway would have been beneficial as a further control.

3. If relating this tool to autophagy-linked events is not the key message of the paper, I might then actually agree with Reviewer 4's opinion in suggesting that the main findings of the manuscript are not novel per se. The authors state that the novelty their manuscript provides lies in:

- Being able to more accurately measure the sub-cellular pH (to a point where subtle changes can be measured that previously wasn't possible)

- Being able to measure the distribution of pH within the cell, rather than just measuring the average pH of the cell

Apart from what shown in Fig. 2, there is no direct comparison of their method with previous FPs, and a thorough comparison of their subcellular behavior(s) and readout(s). Unlike other red FPs engineered to respond to acidic pH (<https://www.nature.com/articles/s41467-021-21666-7>), mApple has been created and characterized previously. The drugs used in the manuscript also are well known. The use of Stardust has also been previously described, therefore DL does not substantiate the novelty here, and same thing goes for the use of FLIM to report on pH variations.

What is the key advantage of their tool and approach?

1. How are the end-users capable of discriminating between this tool/approach and the already-existing ones? On what basis should they make a choice of what is the best FP? Recently, mScarlet has also been shown to change its lifetime in response to pH variations

(https://andrewgyork.github.io/mScarlet_lifetime_reports_pH/), therefore the collection of FPs changing their lifetime behavior upon lifetime variation is rapidly increasing. Although such a comparison with mScarlet is not expected at this stage, key data (BafA1, chloroquine, Stardust analyses...) obtained with mApple should at least be compared to similar experiments obtained with previously-characterized FPs.

2. What is the advantage of measuring the subcellular pH and its distribution? What can their tool do that others can't? What answer does that provide that the others can't? Do we need a tool reporting on <0.1 pH uncertainty in terms of biological relevance? So far, what I see in the manuscript is a methodological paper recollecting previously-existing tools and approaches, while no intrinsic advantage of pHLIM and mApple are presented.

Reviewer #3 (Remarks to the Author):

The authors propose an elegant method based on phasor-FLIM to estimate the pH in individual vesicles. They claim the method developed might be employed in other biological systems.

The authors answered all the reviewer's comments with sound argumentations and additional experiments. The manuscript would be complete if all the fluorescence images would report a color bar indicating the number of photons/pixel dwell time. This would make the interpretation of the images simpler as some of them (especially the intensity images representing fluorescent mApple, like top row Fig2, but not only) appear saturated.

No major revisions are asked.

Reviewer #1

The authors performed part of the suggested experiments to draw more reliable conclusions regarding the proton sponge effect. In the revised manuscript they looked at the pH of endosomes containing actual pDNA/PEI complexes and on the population level found no difference with the pH in endosomes not containing complexes. It is, however, important to realize that the analysis was done on a population level, i.e. averages were calculated which were not significantly different. This is where the problem lies regarding the conclusions that are drawn.

The reviewer is correct that the average pH quoted is a population level mean, however we think they have missed that pHLM also shows the distribution of pH from individual endosomes within the population (see histograms in Figs 3e,f, 4c and 5c,e). This gives significantly more information about pH behaviour inside the cell than the simple population mean.

More specifically, Fig 5e compares pH of endosomes that contain PEI/DNA to the pH of endosomes in the same cell that do not contain PEI/DNA. The mean pH and the distribution of pH of endosomes with and without PEI is not significantly different. We have revised the manuscript as follows to make this point clearer.

“By using Cy5 labelled PEI/DNA complexes, we were also able to measure the pH of individual PEI/DNA positive endosomes, and compare the pH to endosomes in the same cell without PEI/DNA (Fig 5e). There was no difference in the mean pH or the pH distribution in TMEM106b-mApple vesicles with or without PEI.”

To further help clarify that the analysis is performed on individual endosomes, we have changed how we refer to the vesicle pH throughout the manuscript. We realise that by referring to the “mean weighted pH” of each vesicle, it could be interpreted that this is a mean pH of all vesicles, rather than being the mean weighted pH of each individual pixel within the vesicle. We have changed all instances of “mean weighted pH” to simply “pH”, and included the following explanation about how the vesicle pH is calculated.

“This algorithm calculates the endosomal pH by interrogating each pixel in the detected endosome and calculating the intensity weighted mean G value (equation 3). The G value is converted to pH using the calibration curve in Fig. 1h.”

We have also revised the figure captions to make it clearer that the data shows the pH distribution of individual endosomes, and not just a population level mean.

“The pH of each individual endosome is plotted in the histogram with the mean pH of the population shown as a dotted line.”

Endosomal escape by PEI complexes is known to be highly inefficient in the sense that at best a few percentage of endocytosed complexes actually manage to escape.

We agree with this statement, and have added the following sentence to make this point clearer.

“Delivery to the cytosol (also referred to as endosomal escape⁴²) is very inefficient, with less than 2% of internalised material being trafficked to the cytosol^{43,44}.”

This has been attributed to variability in PEI content per endosome and the fact that endosomes can have different sizes. Indeed, to induce a sufficient amount of osmotic pressure through the proton sponge mechanism, there obviously should be a sufficient amount of polymer present in a particular endosome. Apparently this condition is met in only a few percentage of cases (i.e. polyplex containing endosomes). Therefore, it is reasonable to expect that the buffering action will only be present in a few percent of endosomes, and this may be obscured when analysis is done on averages.

We thank the reviewer for raising this important point. They are correct that the amount of PEI per endosome could influence the potential pH buffering of the endosome. This highlights a significant advantage of our technique, which is that we can correlate the intensity of PEI in each endosome with the pH of the endosome. We have performed this additional analysis and included it in the revised version of the manuscript (Figure 5f and Figure S23). We have revised the manuscript as follows to describe the additional analysis:

“We further investigated to see if there was a correlation between the amount of PEI in each endosome (measured from the Cy5 intensity) and the pH of the endosome. Increased sequestration of PEI in endosomes does not correlate with increased endosomal pH (Fig. 5f and Fig. S23).”

This analysis shows that higher amounts of PEI do not correlate with a higher endosomal pH. Combined with Fig. 5e, which shows the mean pH and the distribution of pH in endosomes with and without PEI is not significantly different, we believe this comprehensively supports our conclusion that the proton sponge effect is not the mechanism by which PEI promotes endosomal escape.

We disagree with the reviewer that the buffering action would only be observed in a few percent of endosomes. We would not expect a binary yes/no buffering effect once a threshold concentration of PEI is reached. Buffering is a concentration dependant process, therefore we would expect a correlation between PEI concentration and buffering capacity.

Figure 5f shows the correlation between PEI concentration and endosomal pH for >2500 individual endosomes from 3 independent experiments. There is no correlation between PEI concentration and endosomal pH. There is also no population of endosomes with higher pH. We wish to reiterate that this analysis is not done on averages, but by comparing every individual endosome.

We have revised the manuscript as follows to make these point clearer:

“By using Cy5 labelled PEI/DNA complexes, we were also able to measure the pH of individual PEI/DNA positive endosomes, and compare the pH to endosomes in the same cell without PEI/DNA (Fig 5e). There was no difference in the mean pH or the pH distribution in TMEM106b-mApple vesicles with or without PEI. Furthermore, in addition to measuring the pH of each individual endosome, we measured the fluorescence intensity of Cy5 in each endosome to determine the relative amount of PEI. We would anticipate that if PEI exerts a buffering effect in endocytic vesicles, vesicles with a greater amount of PEI would have a higher pH. By plotting the amount of PEI vs pH for >2500 individual endosomes from 3 independent replicates (Fig. 5f) we have shown that increased sequestration of PEI in endosomes does not correspond to a higher endosomal pH.”

This is also exactly why I had proposed to look at endosomal escape events, because only then you can draw conclusions whether buffering is related to the escape events. The authors decided not to do this, and I will not insist on it since it's not the major point of the paper. Nevertheless I do insist on a more thoughtful discussion, including the above mentioned limitations of the performed analysis, before drawing the in my opinion oversimplified conclusion that buffering is not part of the proton sponge effect.

We thank the reviewer for recognising that detecting endosomal escape events is not a major point of the paper. We have revised the discussion of the proton sponge effect to reflect the additional analysis that we have performed, which further strengthens our conclusion that the proton sponge effect is not the driving force behind PEI induced endosomal escape.

“Our results here show that a) the average pH of vesicles does not change with PEI treatment, b) there is no population of vesicles with higher pH and the distribution of pH is similar regardless of if the vesicle contains PEI or not, c) there is no correlation between the amount of PEI in the vesicle and the pH. All combined, this strongly suggests that the proton sponge effect is not the predominant mechanism by which cytosolic delivery is induced by PEI.”

Reviewer #2:

First of all, I'd like to sincerely congratulate the authors for the impressive experimental work done during the revision phase. The manuscript has greatly improved, and the characterization of the properties of mApple in response to pH is now more careful. The conclusions they reach are more supported by data, resulting in an overall amelioration of the manuscript. I would also like to thank them for taking the time to explain their reasoning, both to me and in the manuscript. I feel that the reading of the manuscript is now easier for the general audience and to me, as a potential end-user of their tool.

We thank the reviewer for recognising the additional experiments we have performed and for their suggestions to improve the manuscript.

I have very few comments, which can be found here below. However, the explanations provided in the response file reinforced the impression that I don't see a clear advantage of this tool and approaches presented in the manuscript, compared to the ones already existing in the literature.

1. As far as I understand, Fig S13 reports on data obtained with a purified mApple protein. If that is the case, please consider adding that it is the purified protein we're looking at, and not at cells expressing a vector coding for mApple alone.

The reviewer is correct that the data generated in Fig S13 (Now S15 in the revised manuscript) was obtained with purified mApple protein. We have revised the figure legend to clarify this point.

“Supplementary figure 15. Effect of BafA and Chloroquine on the lifetime of mApple. Recombinant mApple protein was incubated with 100nM BafA or 100µM of Chloroquine for 45 min prior to imaging.

The addition of either chemical did not affect the pH measurement inferred from mApple fluorescent lifetime.”

2. Point 8: the effect of BafA1 in blocking the autophagy flux is not my hypothesis, but it has been experimentally verified. Please refer to <https://doi.org/10.1080/15548627.2020.1797280> for further details. In this light, the effect of BafA1 the authors observe on the overall vesicle number is very surprising and against what previously shown, and I wonder whether this is linked to a specific feature of the cells they use. In this light, the KO/KD of key genes involved in the autophagosomal pathway would have been beneficial as a further control.

We didn't mean to infer that blocking autophagic flux was just the reviewer's hypothesis. We included references in our previous revision of the manuscript that supported this hypothesis and have revised the manuscript further with additional references to clarify this point.

“BafA1 increases endosomal pH by inhibiting the V-ATPase H⁺ pump³⁵, which prevents acidification of vesicles. It has also been shown that BafA1 inhibits the SERCA Ca²⁺ pump, which disrupts autophagosome/lysosomal fusion independently of its effect on lysosomal pH^{39,40}. However, it is possible the disruption of lysosomal fusion could play a role in increasing the pH of endosomes. To investigate this further we analysed the distribution of vesicle pH and the number of vesicles detected throughout the BafA1 treatment. If V-ATPase H⁺ pump inhibition is the primary mechanism for increasing the endosomal pH, we would expect to see a steady increase in the pH of all the endocytic vesicles. However, if disruption of autophagosome/lysosomal fusion prevents acidification of the vesicles, we would expect to observe two vesicle populations; the initial population of vesicles with lower pH; and a new population of vesicles with a higher pH that are unable to fuse to the autophagosome/lysosomal compartments.”

We disagree with the reviewer that KO/KD studies will be beneficial to support the conclusions in this manuscript. We are not attempting to suggest that BafA1 does not block autophagic flux. Our conclusion is that the change in pH induced by BafA1 is due to its effect on V-ATPase, and not due to the fact that BafA1 also blocks autophagosomal/lysosomal fusion. This work does not contradict the original paper that demonstrated that BafA1 blocks autophagosomal/lysosomal fusion. The conclusion from this paper was that “BafilomycinA1 inhibits fusion independent of its effect on lysosomal pH”.

To make this point clearer we have revised the manuscript as follows.

“This shows that the primary mechanism of BafA1 induced lysosomal neutralisation is inhibition of V-ATPase H⁺ pumps. It should be noted that this result does not contradict the findings that BafA1 can also inhibit autophagosome/lysosomal fusion. However, the lack of a second population of vesicles with a higher pH and the consistent number of vesicles suggests that over the 60 min time course of the experiment, inhibiting autophagosomal/lysosomal fusion is not the driving force behind neutralisation of TMEM106b+ vesicles.”

3. If relating this tool to autophagy-linked events is not the key message of the paper, I might then actually agree with Reviewer 4's opinion in suggesting that the main findings of the manuscript are not novel per se. The authors state that the novelty their manuscript provides lies in:

- Being able to more accurately measure the sub-cellular pH (to a point where subtle changes can be measured that previously wasn't possible)
- Being able to measure the distribution of pH within the cell, rather than just measuring the average pH of the cell
Apart from what shown in Fig. 2, there is no direct comparison of their method with previous FPs, and a thorough comparison of their subcellular behavior(s) and readout(s). Unlike other red FPs engineered to respond to acidic pH (<https://www.nature.com/articles/s41467-021-21666-7>), mApple has been created and characterized previously.

The reviewer may have missed the additional data we provided in the supporting information that shows a direct comparison of mApple to 3 other fluorescent proteins (Figure S2). We included a comparison to muGFP (a pH sensitive protein), pHluorin (a protein specifically engineered to pH measurements) and mCherry (a pH insensitive protein). To further demonstrate the comparison of mApple to these proteins we have also included uncertainty analysis of the pH which can be inferred from these proteins (Fig. S3). We have also expanded our discussion in the main text to compare the advantages of mApple over the other fluorescent proteins.

“It is important to note that the uncertainty in the inferred pH comes from both the intensity/lifetime measurement, and the pH calibration. This latter source of uncertainty is often ignored, but can significantly affect the accuracy of the measurement. When modelling the sigmoidal response of intensity to pH, the exponential and asymptotic regions of the curve have substantially higher uncertainty than the linear region. This is exemplified by the interpolation of pH from the intensity of muGFP, pHluorin and mApple (Fig S3). For each of these fluorophores, the uncertainty in pH is substantially higher than if the lifetime of mApple is used for pH calibration.”

The reviewer is correct that mApple has been created and characterized previously. However, it has not been used to quantify pH. Papers such as the referenced pHmScarlet are certainly interesting developments in the field, however we note that the pH sensitive range of pHmScarlet is 6-8, which does not cover the whole physiologically relevant pH range. It does not enable the sensing of low pH vesicles. Furthermore, it also suffers from the same limitations of all ratiometric pH sensors, as we have highlighted in Fig. 1, S2 and S3.

We contend that identifying properties of an existing protein that enables more accurate pH determination over the full physiologically relevant range is as (or even more) significant than developing a novel protein that has more limited sensing properties.

The drugs used in the manuscript also are well known.

We agree that the drugs in the manuscript are well known, and were chosen because their function is well known. However, as outlined in Figs. 4 and 5, our pHLIM approach provides new insight into the mechanism of both BafA1 and PEI.

The use of Stardust has also been previously described, therefore DL does not substantiate the novelty here, and same thing goes for the use of FLIM to report on pH variations. What is the key advantage of their tool and approach?

While deep learning with Stardist is an established technique, it has not been used to identify endosomes, nor has it been used to measure pH. The novelty of our approach is

- 1) Simplicity – our technique requires a single fluorophore and single measurement, rather than generating two fused fluorophores and making ratiometric measurements of each fluorophore.
- 2) Accuracy – our pH measurements are accurate to <0.1 pH unit, compared to >0.5 for intensity based measurements
- 3) Responsive range – our pH measurements show a linear response across the entire pH range, compared to sensors like pHluorin that cannot sense pH changes below pH 6.
- 4) Quantifying individual endosomes – our method determines the distribution of pH within the cell, enabling small changes in subsets of endosomes to be detected
- 5) Correlating endosome pH with endosome composition – we are able to identify which endosomes contain material (such as PEI) and correlate the pH to the concentration of material in the endosome.

The combination of all these aspects makes our work highly novel and innovative. We have revised the conclusions of the paper to make these advantages clearer.

“We have demonstrated that FLIM measurements of mApple, combined with automated analysis of individual endosomes enables quantitative and accurate measurement of intracellular pH across the physiologically relevant pH range. This technique has a number of advantages over existing methods.

1) Simplicity: FLIM only requires a single measurement, rather than needing ratiometric measurements of two fluorophores. 2) Accuracy: our pHLIM measurements are accurate to <0.1 pH unit, compared to >0.5 for intensity-based measurements. 3) Responsive range: mApple exhibits a linear lifetime response across the entire physiological pH range. 4) Sub-cellular quantification: the application of StarDist enables the distribution of pH within the cell to be determined. 5) Endosome composition: we can identify which endosomes contain material (such as PEI) and correlate the pH to the amount of material in the endosome.”

1. How are the end-users capable of discriminating between this tool/approach and the already-existing ones? On what basis should they make a choice of what is the best FP? Recently, mScarlet has also been shown to change its lifetime in response to pH variations (https://andrewgyork.github.io/mScarlet_lifetime_reports_pH/), therefore the collection of FPs changing their lifetime behavior upon lifetime variation is rapidly increasing. Although such a comparison with mScarlet is not expected at this stage, key data (BafA1, chloroquine, Stardust analyses...) obtained with mApple should at least be compared to similar experiments obtained with previously-characterized FPs.

The data shown on the github webpage shows that the pH response for mScarlet exhibits a sigmoidal response from pH 4 to 7.5, which as outlined in Fig. 2c,d and S3 of our manuscript leads to a high degree of uncertainty than if the lifetime measurement is used. We have revised the manuscript as follows to highlight the importance of a linear response to accurately infer the sub cellular pH.

“When modelling the sigmoidal response of intensity to pH, the exponential and asymptotic regions of the curve have substantially higher uncertainty than the linear region. This is exemplified by the interpolation of pH from the intensity of muGFP, pHluorin and mApple (Fig S3).”

We note that the github webpage was published after we submitted this manuscript and is not peer-reviewed. For these reasons we have not referenced this work in the manuscript, although it does speak to the need and interest for improved subcellular pH sensors.

2. What is the advantage of measuring the subcellular pH and its distribution? What can their tool do that others can't? What answer does that provide that the others can't?

Do we need a tool reporting on <0.1 pH uncertainty in terms of biological relevance?

We thank the reviewer for highlighting the need to better explain why reporting pH with lower uncertainty is important. We have revised the manuscript as follows to make this clearer.

"The increased accuracy of the pH measurements using pHLIM means we can identify pH changes with a high degree of certainty as soon as subtle changes occur (as per treatment with BafA1 – Fig 4) and definitively demonstrate when no changes occur (as per treatment with PEI – Fig 5)."

So far, what I see in the manuscript is a methodological paper recollecting previously-existing tools and approaches, while no intrinsic advantage of pHLIM and mApple are presented.

As per our response to the points above, we have revised the conclusions of the paper to explain the advantages of pHLIM.

Reviewer #3

The authors propose an elegant method based on phasor-FLIM to estimate the pH in individual vesicles. They claim the method developed might be employed in other biological systems. The authors answered all the reviewer's comments with sound argumentations and additional experiments. The manuscript would be complete if all the fluorescence images would report a color bar indicating the number of photons/pixel dwell time. This would make the interpretation of the images simpler as some of them (especially the intensity images representing fluorescent mApple, like top row Fig2, but not only) appear saturated. No major revisions are asked.

We thank the reviewer for this suggestion. We have included Figs S7 and S8 to include a scale which shows the photons per pixel for the intensity based images.

REVIEWERS' COMMENTS

Reviewer #1 (Remarks to the Author):

I thank the authors for further considering my remarks and performing further analysis on the correlation between PEI content and extent of buffering per endosome. I agree that the data now better supports the conclusions that are drawn. Thank you for the clarifications and congratulations on very nice work.

Reviewer #2 (Remarks to the Author):

I have no further question for the authors